# REGIONED EPISODIC REINFORCEMENT LEARNING

## ABSTRACT

Goal-oriented reinforcement learning algorithms are often good at exploration, not exploitation, while episodic algorithms excel at exploitation, not exploration. As a result, neither of these approaches alone can lead to a sample-efficient algorithm in complex environments with high dimensional state space and delayed rewards. Motivated by these observations and shortcomings, in this paper, we introduce Regioned Episodic Reinforcement Learning (RERL) that combines the episodic and goal-oriented learning strengths and leads to a more sample efficient and effective algorithm. RERL achieves this by decomposing the space into several sub-space regions and constructing regions that lead to more effective exploration and high values trajectories. Extensive experiments on various benchmark tasks show that RERL outperforms existing methods in terms of sample efficiency and final rewards.

## 1 INTRODUCTION

Despite its notable success, the application of reinforcement learning (RL) still suffers from sample efficiency in real-world applications. To achieve human-level performance, episodic RL (Pritzel et al., 2017; Lee et al., 2019) is proposed to construct episodic memory, enabling the agent to assimilate new experiences and act upon them rapidly. While episodic algorithms work well for tasks where it is easy to collect valuable trajectories and easy to design dense reward functions, both of these requirements become roadblocks when applying to complex environments with sparse reward. Goal-oriented RL (Andrychowicz et al., 2017; Paul et al., 2019) decomposes the task into several goal-conditioned tasks, where the intrinsic reward is defined as the success probability of reaching each goal by the current policy and the ability to guide the agent to finally reach the target state. These methods intend to explore more unique trajectories and use all trajectories in the training procedure, which may involve unrelated ones and result in inefficient exploitation. In this paper, we propose a novel framework that can combine the strengths of episodic and goal-oriented algorithms and thus can efficiently explore and rapidly exploit high-value trajectories.

The inefficient learning of deep RL has several plausible explanations. In this work, we focus on addressing these challenges: (**C1**) Environments with a sparse reward signal can be difficult to learn, as there may be very few instances where the reward is non-zero. Goal-oriented RL can mitigate this issue by building intrinsic reward signals (Ren et al., 2019), but suffer from the difficulty of generating appropriate goals from high-dimensional space. (**C2**) Training goal-oriented RL models using all historical trajectories rather than selected ones would involve unrelated trajectories in training. The training process of goal generation algorithms could be unstable and inefficient (Kumar et al., 2019), as data distribution shifts when the goal changes. It can be fairly efficient if updates happen only with highly related trajectories. (**C3**) Redundant exploration is another issue that limits the performance as it is inefficient for the agent to explore the same areas repeatedly (Ostrovski et al., 2017). Instead, it would be much more sensible for agents to learn to divide the task into several sub-tasks to avoid redundant exploration.

In this paper, we propose Regioned Episodic Reinforcement Learning (RERL), which tackles the limitations of deep RL listed above and demonstrates dramatic improvements in a wide range of environments. Our work is, in part, inspired by studies on psychology and cognitive neuroscience (Lengyel & Dayan, 2008; Manns et al., 2003), which discovers that when we observe an event, we scan through our corresponding memory storing this kind of events and seek experiences related to

this one. Our agent regionalizes the historical trajectories into several region-based memories*. At each timestep, the region controller will evaluate each region and select one for further exploration and exploitation. Each memory binds a specific goal and a series of goal-oriented trajectories and uses a value-based look-up to retrieve highly related and high-quality trajectories when updating the value function. We adopt hindsight (i.e., the goal state is always generated from visited states in the memory) and diversity (i.e., goal state should be distant from previous goal states in other memories) constraints in goal generation for goal reachability and agent exploration. This architecture conveys several benefits: (1) We can automatically construct region-based memory by goal-oriented exploration, where trajectories guided by the same goal share one memory (see Section 3.1). (2) Within each memory, we alleviate the high-dimensional issue (**C1**) by enforcing that the goal space is a set of visited states (see Section 3.2). (3) In order to improve efficiency in exploitation (**C2**), our architecture stabilizes training using trajectories within the memory instead of randomly selected transitions (see Section 3.3 for details). (4) Our algorithm takes previous goals in other memories when generating a goal in current memory. Specifically, we propose the diversity constraint to encourage the agent to explore unknown states (see Section 3.2), which aims at improving exploration efficiency (**C3**).

The contributions of this paper are as follows: (1) We introduce RERL, a novel framework that combines the strengths of episodic RL and goal-oriented RL for efficient exploration and exploitation. (2) We propose hindsight and diversity constraints in goal generation, which allows the agents to construct and update the regioned memories automatically. (3) We evaluate RERL in challenging robotic environments and show that our method can naturally handle sparse reward environments without any additional prior knowledge and manually modified reward function. RERL can be closely incorporated with various policy networks such as deep deterministic policy gradient (DDPG (Lillicrap et al., 2015)) and proximal policy optimization (PPO (Schulman et al., 2017)). Further, ablation studies demonstrate that our exploration strategy is robust across a wide set of hyper-parameters.

## 2 PRELIMINARIES

In RL (Sutton & Barto, 2018), the goal of an agent is to maximize its expected cumulative reward by interacting with a given environment. The RL problem can be formulated as a Markov Decision Process (MDP) by a tuple $(\mathcal{S}, \mathcal{A}, \mathcal{P}, r, \gamma)$, where $\mathcal{S}$ is the state space, $\mathcal{A}$ is the action space, $\mathcal{P} : \mathcal{S} \times \mathcal{A} \to \Delta(\mathcal{S})$ is the state transition probability distribution, $r : \mathcal{S} \times \mathcal{A} \to [0, 1]$ is the reward function, and $\gamma \in [0, 1)$ is the discount factor for future rewards. Our objective is to find a stochastic policy $\pi : \mathcal{S} \times \mathcal{A} \to [0, 1)$ that maximizes the expected cumulative reward $R_t = \sum_{k=0}^{T} \gamma^k r_{t+k}$ within the MDP, where $T$ is the episode length. In the finite-horizon setting, the state-action value function $Q^\pi(s, a) = \mathbb{E}[R_t | s_t = s, a]$ is the expected return for executing action $a$ on state $s$ and following $\pi$ afterward. The value function can be defined as

$$V^\pi(s) := \mathbb{E}\left[\sum_{k=0}^{T} \gamma^k r_{t+k}(s_t, a_t) \mid s_t = s, \pi\right], \forall s \in S, \tag{1}$$

where $T$ is the episode length and the goal of the agent is to maximize the expected return of each state $s_t$. Deep $Q$ Network (DQN, (Mnih et al., 2015)) utilizes an off-policy learning strategy, which samples $(s_t, a_t, r_t, s_{t+1})$ tuples from a replay buffer for training. It is a typical parametric RL method and suffers from sample inefficiency due to slow gradient-based updates. The key idea of episodic RL is to store good past experiences in a tabular-based non-parametric memory and rapidly latch onto past successful policies when encountering similar states, instead of waiting for many optimization steps. However, in environments with sparse rewards, there may be very few instances where the reward is non-zero, making it difficult for an agent to find good past experiences. In order to address this issue, goal-oriented RL is proposed. In the goal-conditioned setting that we use here, the policy and the reward are also conditioned on a goal $g \in \mathcal{G}$ (Schaul et al., 2015). The distance function $d$ (used to define goal completion and generate sparse reward upon the completion of goal) may be exposed as a shaped intrinsic reward without any additional domain knowledge: $r(s_t, a_t | g) = 1$, if $d(\phi(\cdot | s_{t+1}), g) \leq \delta$, and $r(s_t, a_t | g) = -d(\phi(\cdot | s_{t+1}), g)$ otherwise, where $\phi$ :

---

*The common idea our method shares with neuroscience is utilizing highly related information to promote learning efficiency. The difference is that memories are regioned according to the generated goals in this paper, and fictions in cognitive neuroscience.

**Algorithm 1** Framework of RERL

1: **repeat**
2:     Select Region together with Region-based Memory.
3:     Generate goals for exploration with Goal-oriented RL.
4:     Interact with the Environment.
5:     Store historical trajectories into Memory.
6:     Update value estimation for exploitation with Episodic RL.
7: **until** $Q$ function Converges.

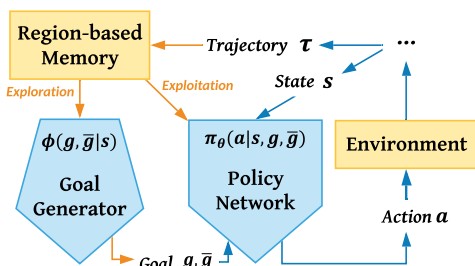

Figure 1: An illustration of RERL where we propose region-based memory for efficient exploration and exploitation.

$\mathcal{S} \to \mathcal{G}$ is a known and tractable mapping[†]. While we expect that cooperation of goal generation and distance function themselves to lead the agent to the final state (global optimum), in practice, we need to consider that there exist local optima due to state space structure or transition dynamics (Trott et al., 2019). Once we can generate appropriate goal $g$ and anti-goal $\bar{g}$, we are able to redefine the intrinsic reward function as:

$$r(s_t, a_t | g, \bar{g}) := \begin{cases} 1 & d(\phi(\cdot|s_{t+1}), g) \le \delta \\ \min[0, -d(\phi(\cdot|s_{t+1}), g) + d(\phi(\cdot|s_{t+1}), \bar{g})] & \text{otherwise,} \end{cases} \quad (2)$$

where $s_{t+1} \sim \mathcal{P}(\cdot|s_t, a_t)$ denotes the next state; $\phi : \mathcal{S} \to \mathcal{G}$ is the extended joint generation for both goal and anti-goal generations; $\bar{g} \in \mathcal{G}$ is the anti-goal and acts as a state that the agent should avoid, which prevents the policy from getting stuck at the local optimum and enables the agent to learn to reach the goal location quickly;(Trott et al., 2019) $\delta$ is a given threshold indicating whether the goal is considered to be reached (Plappert et al., 2018). To make use of $r(s_t, a_t|g, \bar{g})$ in practice, we require a method to dynamically estimate the local optima that frustrate learning without relying on domain-expertise or hand-picked estimations.

The idea of the universal value function (Schaul et al., 2015) is to use a universal functional approximator to represent a large number of value functions. In the goal-oriented scenario, the value function conditioned on any given goal of $g$ and anti-goal $\bar{g}$ can be defined as

$$V^\pi(s, g, \bar{g}) := \mathbb{E}_{a_t \sim \pi(\cdot|s_t, g, \bar{g}), s_{t+1} \sim \mathcal{P}(\cdot|s_t, a_t)} \left[ \sum_{t=1}^{T} \gamma^t r(s_t, a_t|g, \bar{g}) \mid s_t = s \right]. \quad (3)$$

Let $\mathcal{X} : \{x \mid x = (s, g, \bar{g})\}$, denote the joint set over state and goal spaces. Specifically, we define $x* \in \mathcal{X}$ over initial state $s_0 \in \mathcal{S}$, initial goal $g^* \in \mathcal{G}$ and initial anti-goal $\bar{g}^* \in \mathcal{G}$. At the start of every goal-oriented task (Plappert et al., 2018), an initial-terminal states pair will be drawn from the task distribution. In this paper, we regard the terminal state as the original goal $g^*$ and set the original anti-goal $\bar{g}^*$ as the initial state to encourage the agent to explore at the beginning. In this setting, the agent tries to find a policy $\pi$ that maximizes the expectation of discounted cumulative reward $V^\pi(x^*)$. From the comparison of Eqs. (1) and (3), one can see that the critical points for goal-oriented RL are to generate appropriate goals. However, as stated in (Ren et al., 2019), in goal-oriented RL, the value function $V^\pi(x)$ is optimized with respect to a shifting goal-conditioned task distribution, which makes learning unstable. This issue requires RL algorithms to rapidly obtain value estimation under current goal-conditioned tasks, which is the strength of episodic RL. For convenience, we replace all $(s, g, \bar{g})$ tuples with $x$ in the following context.

## 3   REGIONED EPISODIC REINFORCEMENT LEARNING

---

[†]The definition of $\phi$ depends on the definitions of state and goal, and varys when encountering the different environments (Ren et al., 2019). For example, the goal only indicates the designated position of the destination in the Ant Maze environment (see Figure 4(b)), thus, the mapping is defined as a mapping from a system state to the position of the destination in this case.

The basic idea behind this paper is to 'divide-and-conquer'[‡] exploration and exploitation problems in RL. Firstly, we adopt goal-oriented RL to 'divide' the state space into several regions, where a specific goal identifies each region. We then utilize episodic RL to 'conquer', where we store and learn from highly related and high-quality trajectories in region-based memories. The overall framework is called Region Episodic Reinforcement Learning (RERL).

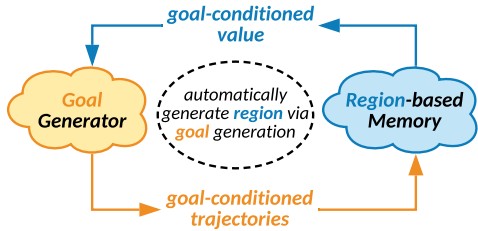

Figure 2: An illustration for motivation.

We provide the overall algorithm in Algorithm 1 and illustration in Figure 1, which combines the strengths of goal-oriented RL and episodic RL to perform efficient exploration (illustrated as orange part in Figure 2) and exploitation (illustrated as blue part in Figure 2). In the following section, we first introduce the definition of regions together with region-based memories in Section 3.1. In order to automatically obtain regions during exploration, we generate appropriate goals to guide the agent under hindsight and diversity constraints in Section 3.2. Since the goal of any RL agent is to learn a policy that can maximize the expected return, we show the value estimation update formulation based on region-based memory in Section 3.3.

## 3.1 CONSTRUCT REGION-BASED MEMORY

Following (Florensa et al., 2017), many previous goal-oriented RL works (Ren et al., 2019; Asadi et al., 2018) adopt Assumption 1 to guarantees continuous goal-space representation.

**Assumption 1.** *A value function $V^\pi(x)$ has Lipschitz continuity over goal $g$ and anti-goal $\bar{g}$, which can be formulated as*

$$|V^\pi(x) - V^\pi(x')| \le L \cdot d(x, x'),\tag{4}$$

where $L$ denotes the Lipschitz constant. Considering that this Lipschitz continuity may not hold for every $x \in \mathcal{X}$, we partition the joint set $\mathcal{X}$ into several subsets. If $d(x, x')$ is not too large within each sub-set, generally speaking, it is reasonable to claim that the bound Eq. (4) holds for most scenarios. In this paper, we define these sub-spaces as regions. We formulate the definition as follows:

**Definition 1.** *Considering that $\mathcal{X}$ satisfies $\mathcal{X} = \bigcup_{i=1}^{N} \mathcal{X}_i$ and $\mathcal{X}_i \bigcap \mathcal{X}_j = \emptyset$, $\forall i, j = 1, 2, \ldots, N$ and $i \ne j$, we define each subset $\mathcal{X}_i$ as a region, where $N$ is the number of regions.*

An ideal partition strategy should divide state space into several parts, and each part leads to one meaningful goal (e.g., in the case of exploring a large house, ideal partition strategy should divide the house into separated rooms). RL algorithms explore each partition while ignoring other state space, thus significantly reducing exploration complexity. However, one should note that it is impractical to find the perfect partition strategy without any task-specific manual engineering. One possible solution to automatically generate these regions is to bind each region with a series of goals. In other words, we can design a region-based goal generation where at each timestep, we pick up one region and update the goal within the region. This architecture conveys several advantages: (1) It allows the agent to solve a complex environment through 'divide-and-conquer'. (2) Goal generation is modified within a sub-space, which can improve the stability.

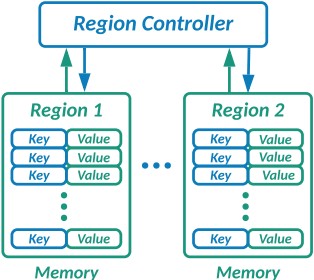

Figure 3: An illustration for region-based memory.

In order to achieve this, we construct region-based memories based on historical trajectories. Specifically, for each region-based memory $\mathcal{M}_n$, we have a simple memory module $M_n(x) = (K(x), V^\pi(x))$, where $x \in \mathcal{X}_n$, $K(x) = (\phi(\cdot|s), g, \bar{g})$ is the key of the memory and $V^\pi(x)$ is the value of the memory. As shown in Figure 8, each memory binds a specific region. At each episode, the region controller selects the region-based memory containing the highest value state for further exploration and exploitation. The motivation behind this is very intuitive that the agent always focuses on the region with the highest potential. However, directly adopting this greedy operation may lead to the phenomena of rich-get-richer (Salganik et al., 2006). Instead, we adopt

---

[‡]Different from traditional divide-and-conquer algorithms, we 'divide and conquer' the problem with only one round of problem division instead of using a recursive way.

Boltzmann softmax to select one region $\mathcal{X}_n$. We use $\mathcal{X}_n$ to denote a division of joint set $\mathcal{X}$, which is conceptual and not accessible. In the practice, we use the trajectories stored in $\mathcal{M}_n$ instead.

$$\text{Selected-}\mathcal{M}_n = \frac{\exp(\max_{m \in \mathcal{M}_n} V_m / \iota)}{\sum_{i=1}^{N} \exp(\max_{m \in \mathcal{M}_i} V_m / \iota)}, \tag{5}$$

where $\iota$ denotes the temperature hyper-parameter to control the exploration rate, $V_m$ is the value of the sampled memory $m$, and $N$ is the number of regions. In practice, we set the initial temperature at 1.0, then gradually reduce the temperature to 0.01 to limit region level exploration. After selecting a region $\mathcal{X}_n$, the agent will focus on performing efficient exploration and exploitation upon the historical experience in its associated memory $\mathcal{M}_n$. We here prove that the value optimization problem in a region-based setting is a relaxed lower bound for the original one through Proposition 1.

**Proposition 1.** *Given the joint set $\mathcal{X}$ and several region-based sets (i.e., sub-sets) $\mathcal{X}_n$, where $n = 1, 2, \ldots, N$ and $N$ is the number of regions, we have*

$$\forall \pi, \ \max_{x \in \mathcal{X}} V^\pi(x) \geq \max_{x \in \{x_1, x_2 \ldots, x_N\}} V^\pi(x), \ where \ x_n = \arg\max_{x_n \in \mathcal{X}_n} V^\pi(x_n). \tag{6}$$

*Proof.* The proof of Proposition 1 is provided in Appendix C.1. □

### 3.2 EXPLORE WITH GOAL-ORIENTED REINFORCEMENT LEARNING

In this section, we aim to find appropriate goals for exploration. In this paper, we analyze that appropriate goals should have the following three properties, namely (1) high value (close to terminal state), (2) reachability (appropriate for current policy), and (3) exploratory potential (explore unvisited states). To this end, we search for high-value states, according to Eq. (3), under hindsight and diversity constraints. Based on Assumption 1, we can easily derive that

$$\forall x_n \in \mathcal{X}_n, x_n' \in \mathcal{X}_n', \ V^\pi(x_n) \geq V^\pi(x_n') - L \cdot d(x_n, x_n'). \tag{7}$$

Jointly considering Eqs. (9) and (7), optimizing cumulative rewards in Eq. (3) can be relaxed into the following surrogate problem:

$$\max_{\pi, x \in \{x_1, x_2, \ldots, x_N\}} V^\pi(x), \ where \ x_n = \arg\max_{x_n \in \mathcal{X}_n}\{V^\pi(x_n) - L \cdot d(x_n, x^*)\}, \ n = 1, 2, 3, \ldots, N, \tag{8}$$

Note that this new objective function is intuitive. Instead of directly optimizing with $x^*$, which is likely to be hard, we hope to find a collection of surrogate sets $x \in \mathcal{X}$, which benefit the exploration, ease the optimization, and are close to or converge towards $x^*$. However, as stated in (Ren et al., 2019), the joint optimization of $\pi$ and $x$ is non-trivial due to high-dimensional observation and shifting distribution during optimization. In order to find appropriate states for goal generation and make the system stable, we then introduce two constraints, namely hindsight constraint for reachability and diversity constraint for exploratory potential.

**Hindsight Constraint.** In order to guarantee goal reachability and improve learning stability, we adopt the idea of hindsight goals (Andrychowicz et al., 2017), which means $\mathcal{G} \subseteq \mathcal{S}$. We first enforce $\mathcal{X}$ on a finite set of $Z$ particles that can only be from those already achieved states from trajectories $\{\tau\}$ in the current memory $\mathcal{M}_k$, which means that the support of $\mathcal{X}$ should be base on $\mathcal{M}_n$.

Deep $Q$ Network (DQN, (Mnih et al., 2015)) parameterizes the action-value function by deep neural networks $Q_\theta(s, a)$ using $Q$-learning (Watkins & Dayan, 1992) to learn which action is the best to take at the timestep $t$. According to Eq. (8), one can see that we are aiming to find high-value states with similar goal-conditioned tasks. Based on the components of region-based memories, we rank and select top-$Z$ trajectories $\{\tau_z\}_{z=1}^{Z}$, where $\tau_z = \{s_t^z\}$ corresponding to goal-oriented task $x_z$, to maximize $\sum_{z=1}^{Z} w(x_z, \tau_z)$, where $w(x_z, \tau_z)$ is defined as

$$w(x_z, \tau_z) := \alpha \, d(x_z, x^*) + \min_{s_t \in \tau_z} \left( \|\phi(\cdot|s_t) - g^*\| - \frac{1}{L} V^\pi(x_z) \right), \tag{9}$$

where the first term is to measure the goal-conditioned task similarity with the key in the memory, and the second term is to select high-value states, and $\alpha$ is the hyperparameter to balance these two terms.

**Diversity Constraint.** In order to encourage the agent to explore unvisited states and avoid the overlapping among the regions, we adopt the diversity constraint in goal generation. Then, we can re-formulate Eq. (9) as

$$
w(x_z, \tau_z) := \alpha \, d(x_z, x^*) + \min_{s_t \in \tau_z} \left( \|\phi(\cdot|s_t) - g^*\| - \frac{1}{L} V^\pi(x_z) - \frac{1}{\beta} \frac{1}{N} \sum_{j \in -n} (\|\phi(\cdot|s_t) - g^j\|) \right),
$$
(10)

where $\beta$ adjusts the weight of the diversity constraint, and $-n$ denotes the set of index except $n$. The motivation behind this is that considering that goals in goal-oriented RL indicate the direction for exploration, the generated goal is expected to be different from historical goals in other regions. Therefore, the formulation of our goal generation can be easily derived from Eq. (10), which can be formulated as

$$
g = \phi \left( \cdot \mid \underset{s_t \in \tau_z}{\arg\min}(\|\phi(\cdot|s_t) - g^*\| - \frac{1}{L} V^\pi(x_z) - \frac{1}{\beta} \frac{1}{N} \sum_{j \in -n} (\|\phi(\cdot|s_t) - g^j\|)) \right),
$$
(11)

where $\tau_z$ is obtained through maximizing $\sum_{z=1}^{Z} w(x_z, \tau_z)$, where $w(x_z, \tau_z)$ is defined according to Eq. (10). For the anti-goal generation, we directly assign the visited state with the average value in the region as the anti-goal. The original motivation for the anti-goal setting is to avoid local optima, which and can be further described as a reward shaping technique (Trott et al., 2019). An illustrated example of the goal generation is shown in Appendix B.1.

### 3.3 EXPLOIT WITH EPISODIC REINFORCEMENT LEARNING

Similar to previous episodic RL algorithms (Lin et al., 2018; Zhu et al., 2019), we adopt region-based memories to maintain the historically highest values $V^\pi(x_t)$ for each joint state-goal distribution and action pair. When encountering a new state, the agent will look up and update the corresponding memory according to the following equation:

$$
V^\pi(x_t) \leftarrow \begin{cases} \max\left(V^\pi(x_t), R_t\right), & \text{if } x_t \text{ satisfies } M_n(x_t) \in \mathcal{M}_n \\ R_t, & \text{otherwise} \end{cases}.
$$
(12)

When the goal is changing ($g \rightarrow g'$), the agent is required to conduct goal relabeling, similar to (Andrychowicz et al., 2017). That is, the agent needs to firstly update the key ($K(x) \rightarrow K(x')$), then re-calculate the reward ($R_t \rightarrow R'_t$) and update the value according to Eq. (12). Note that RERL enables the agent to rapidly assimilate new experiences to improve sample efficiency by looking up the region-based memory. Furthermore, slowly changing goal-conditioned tasks guarantees stability by restricting goal updating within each region. Based on the up-to-date region-based memories, our algorithm can be adapted to various RL training algorithms. We give a proof of convergence in Appendix C.2, our algorithm can converge to a unique optimal point when using $Q$-learning for value learning.

**Overall Algorithm**. We provide the overall algorithm in Algorithm 2 in Appendix 2. We also provide some other views, including curriculum learning and maximum entropy reinforcement learning, to better understand how RERL works. Please refer to Appendix B.2 and B.3 for details.

## 4 EXPERIMENTS

In this section, we perform an experimental evaluation of the proposed method of learning from trajectories and compare it with other state-of-the-art methods. We also perform an ablation study

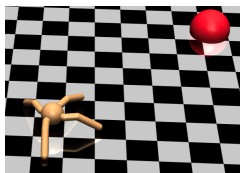 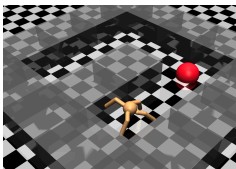 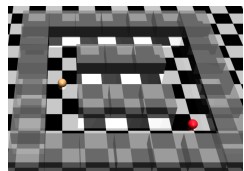 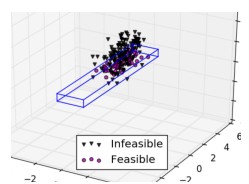

(a) Free Ant Locomotion    (b) Maze Ant Locomotion    (c) Multi-Path Point Mass    (d) *N*-dimension Point Mass

Figure 4: In (a)-(c), the red areas are goals reachable by the orange agent. In (d) any point within the blue frame is a feasible goal (purple balls) and the rest are unfeasible (black triangles).

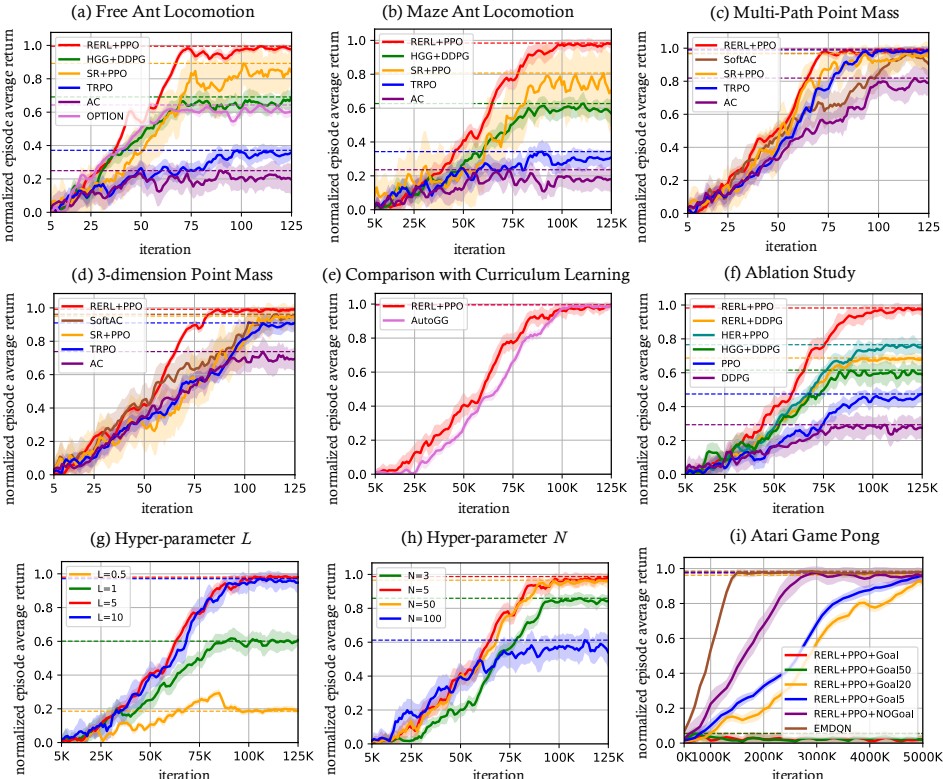

Figure 5: Learning curves of RERL, HGG, HER, SR, AutoGG, EMDQN and POINT on various environments, where the solid curves depict the mean, the shaded areas indicate the standard deviation, and dashed horizontal lines show the asymptotic performance. Several curves are omitted due to page limitations. A full version is deferred to Appendix E.1.

of different settings of our framework. In this section, we provide the experimental results to answer the following questions:

1. Can our RERL approach obtain better convergence in various environments?
2. Can our goal generation enhance the RL method to achieve asymptotic performance with higher efficiency?
3. Can our RERL tackle a complex multi-path goal distribution?
4. Can our RERL scale to higher-dimensional goal-spaces?
5. Do our generated goals really encourage exploration?

To answer the first two questions, we demonstrate our method in two challenging robotic locomotion tasks (see Figure 5(a)(b)). To answer the third question, we train an ant agent to reach any position within a multi-path maze (see Figure 5(c)). To answer the fourth question, we investigate how our method performs with the dimension of goal-space in an environment (see Figure 5(d) for the 3D case). To answer the final question, we conduct a visualization study (see Figure 6) on generated goals. Specifically, we conduct extensive experiments with existing approaches:

- **HER**: Andrychowicz et al. (2017) introduced Hindsight Experience Replay , which constructs imaginary goals in a simple heuristic way to tackle the sparse reward issue.
- **HGG**: Ren et al. (2019) proposed a Hindsight Goal Generation incorporating with DDPG (Lillicrap et al., 2015) that generates valuable hindsight goals to guide the agent.
- **AutoGG**: Florensa et al. (2017) leveraged Least-Squares GAN (Mao et al., 2018) to mimic the set of Goals of Intermediate Difficulty as an automatic goal generator.
- **SR**: Trott et al. (2019) proposed a novel framework named Sibling Rivalry accompanied by PPO Schulman et al. (2017) for learning from sibling trajectories with self-balancing reward.
- **POINT**: Jinnai et al. (2019) proposed to extend covering options to large state spaces, automatically discovering task-agnostic options that encourage exploration.
- **EMDQN**: Lin et al. (2018) leverages episodic memory to supervise an agent during training.

Note that RERL can be closely incorporated with policy networks such as A2C (Mnih et al., 2016), DDPG (Lillicrap et al., 2015), TRPO (Schulman et al., 2015), PPO (Schulman et al., 2017), Soft-AC (Haarnoja et al., 2018), etc. The detailed description of experiment settings and implementation

details can be found in Appendix D.1 and D.3. In this paper, we implement HGG+DDPG and SR+PPO, as initially proposed.

**Ant Locomotion.** We test RERL in two challenging environments of a complex robotic agent navigating either a free space (Free Ant, Figure. 4(a)) or a U-shaped maze (Maze Ant, Figure. 4(b)). Duan et al. (2016) described the task of trying to reach the other end of the U-turn, and they show that standard RL methods are unable to solve it. We further extend the task to evaluate whether the agent is able to reach any given position ($\epsilon$-balls depicted in red) within the maze for Maze Ant or within the target square for Free Ant. As showed in Figure 5(a)(b), the performance of our approach exceeds that of the baselines above.

**Multi-Path Point-Mass Maze.** We show that RERL is efficient at tracking clearly multi-path distributions of goals. To this end, we introduce a new maze environment with multiple paths, as illustrated in Figure 4(c). As in the experiment above, our task is to learn a policy that can reach any feasible center of the mass $(x, y)$ corresponding to $\epsilon$-balls in state space, like the one depicted in red. As shown in Figure 5(c), our approach obtains better performance even in a multi-path environment where goal distribution is naturally more complex than previous environments (see Appendix E.2 for demonstration).

**$N$-dimensional Point-Mass Maze.** We use an $N$-dimensional Point-Mass to demonstrate the performance of our method as the state space dimension increases. As shown in Figure 5(d), our approach outperforms strong baselines in the high-dimensional experiment.

**Atari Game Pong** We evaluate RERL in Atari Game, where several episodic RL algorithms (Badia et al., 2020; Lin et al., 2018) have achieved good performance. In the previous goal-oriented environments such as Maze, both state and goal have physical meaning (e.g., location in the maze). Therefore, it is easy for us to define the distance between two states, which denotes the physical distance in the maze. However, in the Atari Game environment, both state and goal are the image. Hence, the distance here has no physical meaning, which implies that directly attending the goal-oriented setting will result in bad performance. In order to verify the analysis above, we directly use the extrinsic reward from environment (denoted as RERL+PPO+NOGoal), and then gradually (i.e., 5%, 20%, 50%) add the intrinsic reward in the reward function (denoted as RERL+PPO+Goal5, ERL+PPO+Goal20, ERL+PPO+Goal50, respectively). We present the result in Figure 5(i). Results show that the goal-oriented setting of RERL is not suitable for the environment like Atari Game. Also, in the Atari Game environment, there are less sparse rewards than the Ant Maze environment. Therefore, simple episodic RL algorithm such as EMDQN can obtain better performance than RERL.

More experimental results of environments above can be found in Appendix E.1.

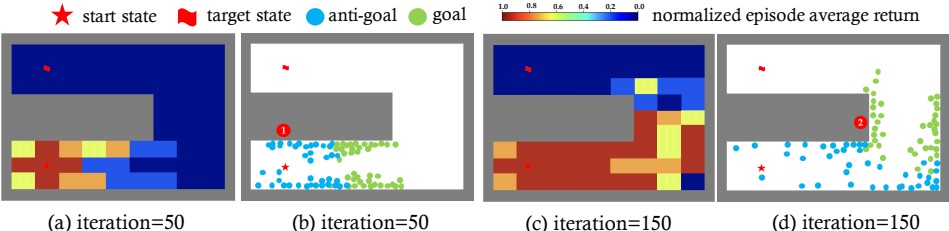

(a) iteration=50          (b) iteration=50          (c) iteration=150          (d) iteration=150

Figure 6: Each grid cell in U-maze is colored according to the expected return success rate when fixing its center as the target state.

**Visualization Study on Generated Goals.** In order to investigate whether the generated goals, served as curriculum in curriculum learning, truly guide the agent to the target state and are at an appropriate difficulty level, we show the distribution of generated goals at different training stages. Results in Figure 6 show that the generated goals are approaching as the training proceeds, and at an appropriate success rate level, where the hindsight constraint helps the agent aim at feasible positions while our diversity constraint encourages the agent to approach the target state. More results of visualization study can be found in Appendix E.2.

**Comparison with Explicit Curriculum Learning.** Since our method can be seen as an explicit curriculum learning for exploration in Appendix B.2, we also compare our method with another recently proposed automatic curriculum learning method for goal-oriented RL. (see Appendix E.3

for the detailed experiment). The result in Figure 5(e) indicates that RERL substantially outperforms this explicit curriculum learning approach even with GOID.

**Impact of Goal Generation.** To further investigate the performance gain from RERL, we design the ablation study on goal generation. We incorporate the goal generator with various RL algorithms and evaluate their performance in the Maze Ant Locomotion environment. Results in Figure 5(f) illustrate that RERL significantly helps the RL method obtain effective and stable performance.

**Impact of Hyper-parameter Selection.** Also, we study the effect of hyper-parameter selection here, i.e., Lipschitz constant $L$, number of regions $N$, number of trajectories $Z$, diversity weight $\alpha$ and hindsight weight $\beta$. We conduct the experiments on the Maze Ant Locomotion environment and report the results in Figure 5(g) and (h). Refer to Appendix E.4 for detailed information.

## 5 RELATED WORK

**Goal-oriented Reinforcement Learning.** Goal-oriented RL allows an agent to learn a goal-conditioned policy, which takes the current state and goal state as the input and predicts a sequence of actions to reach the goal (Florensa et al., 2018; Paul et al., 2019). Recent attempts (Andrychowicz et al., 2017; Ren et al., 2019; Pong et al., 2018) combine off-policy RL algorithms with goal-relabelling to efficiently generate appropriate goals from the visited states and guarantee goal reachability using a hindsight constraint. In this paper, we divide the state space into several regions. Within each region, our algorithm learns a goal-conditioned policy to reach the generated goal. While most previous methods (Wu et al., 2018; Drummond, 2002) perform goal-oriented RL on top of the explored trajectories, our method utilizes a diversity constraint in the goal generation procedure to enhance exploration and automatically structure region-based memory.

**Episodic Reinforcement Learning.** Episode RL is proposed by cognitive studies of episodic memory (Sutherland & Rudy, 1989; Marr et al., 1991; Lengyel & Dayan, 2008) in human decision making (Gilboa & Schmeidler, 1995). Recent works have investigated integrating episodic memory with deep $Q$ networks (DQNs) in non-parametric (Blundell et al., 2016) and parametric (Pritzel et al., 2017) ways. In order to fully utilize the episodic memory, value propagation methods (Hansen et al., 2018; Zhu et al., 2019) have been proposed to obtain trajectory-centric value estimates. A common theme in recent work is finding similar historical trajectories to estimate the value function. While most methods use look-up operations (Pritzel et al., 2017) or graph structure (Zhu et al., 2019), our algorithm explicitly divides trajectories sharing the same goal into one region.

**Hierarchical Reinforcement Learning**. Hierarchical RL learns a set of primitive tasks that together help an agent learn the complex task. There are mainly two lines of work. One class of algorithms (Shang et al., 2019; Nachum et al., 2018; Bacon et al., 2017; Frans et al., 2018; Vezhnevets et al., 2017) jointly learn a low-level policy together with a high-level policy, where the lower-level policy interacts directly with the environment to achieve each task, while the higher-level policy instructs the lower-level policy via high-level actions or goals to sequence these tasks into the complex task. The other class of methods (Drummond, 2002; Fox et al., 2017; Şimşek et al., 2005) focuses on discovering sub-tasks or sub-goals that are easy to reach in a short time and can guide the agent to the terminal state. Recently, several option-discovery approaches (Jinnai et al., 2019; Machado et al., 2017; Bagaria & Konidaris, 2019; Konidaris & Barto, 2009) are proposed to find a set of options to reduce the environment's cover time. Different from these works, the idea of our work is similar to curriculum learning where the sub-tasks are geting harder during the training procedure. As stated by (Nachum et al., 2018), while jointly learning high-level and low-level policies can be unstable, we sidestep the problem by constraining goal generation to be within a specific region under a hindsight constraint.

## 6 CONCLUSION

In this paper, we present a framework that incorporates episodic RL with goal-oriented RL to improve the efficiency of exploration and exploitation. RERL does not require any additional reward engineering or domain expertise. For future work, it would be interesting to further investigate to incorporate our work with representation learning to obtain a better representation of the environment and imitation learning to enhance the learning efficiency with expert knowledge.

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

## A    ALGORITHM

---

**Algorithm 2** Regioned Episodic Reinforcement Learning (RERL)

1: Initialize $\pi$, $g^*$ as terminal state, $\bar{g}^*$ as inital state
2: Initialize region-based memories $\{\mathcal{M}_n\}_{n=1}^N$ by random sample
3: **for** episode $= 1, 2, \ldots, E$ **do**
4:     Select region $n$ according to Eq. (5)
5:     Collect $Z$ trajectories $\{\tau_z\}_{z=1}^Z$ for each $\mathcal{M}_n$ that maximize $\sum_{z=1}^Z w(x_z, \tau_z)$ according to
   Eq. (10)
6:     Construct intermediate goal $g$ according to Eq. (11), $\bar{g}$ for each $\mathcal{M}_n$ with $s$ of average value

7:     **for** $t = 1, 2, \ldots, T$ **do**
8:         $a_t \leftarrow \pi(a|s, g, \bar{g})$
9:         $s_{t+1} \sim \mathcal{P}(\cdot|s_t, a_t)$
10:        $r_t \leftarrow r(s, g, \bar{g})$ according to Eq. (2)
11:        $\mathcal{M}_n \leftarrow \mathcal{M}_n \cup \{M_n(s_t, g, \bar{g}\}$
12:        Update $\mathcal{M}_n$ according to Eq. (12)
13:        Sample a minibatch $b$ from $\mathcal{M}_n$
14:        Update policy $\pi$ on minibatch $b$ using DDPG or PPO
15:    **end for**
16: **end for**

---

The overall description of our algorithm is shown in Algorithm 2. In the initialization procedure, we set the terminal state as the initial goal and initial state as the initial anti-goal, and sample trajectories into each memory. At each episode $e$, the agent selects one region that is most promising to lead to the terminal state in line 4. We construct a goal based on the historical trajectories in line 5. We take previous goals in other memories into consideration in the goal generation in line 6. From line 8 to line 12, the agent interacts with environment and update the memory. Our work focuses on how to build an efficient exploration and exploitation mechanism that is naturally complementary with policy networks such as deep deterministic policy gradient (DDPG (Lillicrap et al., 2015)) and proximal policy optimization (PPO (Schulman et al., 2017)) in line 14.

## B    DICUSSIONS

### B.1    EXAMPLE FOR GOAL GENERATION

Previous works (Florensa et al., 2018; Vezhn-evets et al., 2017) adopt a goal generator to construct immediate intrinsic rewards according to the previous states. However, they often suffer a lot from balancing the efficiency of exploration and exploitation and stability in training. In the first episode, the agent explores two trajectories in different directions, with the closest one $\tau_{1a}$ to the target state labeled as the goal $g_1$ and the farthest one $\tau_{1b}$ as the anti-goal $\bar{g}_1$. In the second episode, the agent evaluates the highest value of states in the regions and selects one according to Eq. (5). The agent does exploration under the guided by $g_1$ (illustrated as sun icon in blue region) and $\bar{g}_1$ (illustrated as moon icon in blue region). If the agent selects the re-

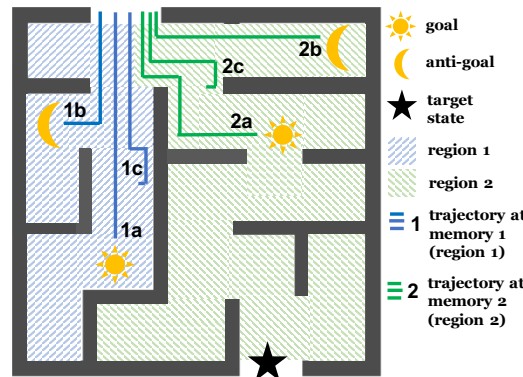

Figure 7: An illustration of exploration strategy.

gion 2, following the similar procedures, the agent will explore the region guided by $g_2$ (illustrated as sun icon in green region) and $\bar{g}_2$ (illustrated as moon icon in green region). Note that goal $g$ will direct the exploration. Hence, in the goal generation, we take the historical goals in the other regions into consideration by the diversity constraint. However, for the anti-goal generation, there is no need to consider other region data as described in Section 3.

## B.2 Relationship to Curriculum Learning

In order to better understand why our method can work in complex environments and can excel other traditional methods more intuitively, we further investigate the relationship between our algorithm from Eq. (3) and empirical utility maximization formulation proposed in (Hacohen & Weinshall, 2019). We provide theoretical analysis that under some assumptions, optimizing our objection function can be similar to optimizing a curriculum algorithm under additional constraints.

Following Section 2, we formulated reinforcement learning problem as a Markov Decision Process (MDP) by a tuple $(\mathcal{S}, \mathcal{A}, \mathcal{P}, r, \gamma)$, where $\mathcal{S}$ is the state space, $\mathcal{A}$ is the action space, $\mathcal{P} : \mathcal{S} \times \mathcal{A} \to \Delta(\mathcal{S})$ is the state transition probability distribution, $r : \mathcal{S} \times \mathcal{A} \to [0,1]$ is the reward function, and $\gamma \in [0,1)$ is the discount factor for future rewards. The utility function is defined as the expected sum of the immediate and long-time utility $U_\pi(s)$ under the policy $\pi : \mathcal{S} \times \mathcal{A} \to [0,1)$, and discount factor $\gamma \in [0,1)$, which can be formulated as:

$$U_\pi(s) := \mathbb{E}_{s_0=s, a_t \sim \pi(\cdot|s_t), s_{t+1} \sim \mathcal{P}(\cdot|s_t, a_t)} \left[ \sum_{t=0}^{T} \gamma^t r(s_t, a_t) \right], \tag{1}$$

where $T$ is the episode length. We can utilize $U_\pi(s)$ to represent the long-time reward (i.e., episode reward). In order to formulate the short-time reward, similarly, we define $U_\pi(s_t)$ by $U_\pi(s_t) := \gamma^t r(s_t, a_t)$. In a similar manner with the Empirical Risk Minimization (ERM) framework, we choose to maximize the average utility, which is defined as follows:

$$\pi^* = \arg\max_\pi \mathcal{U}(\pi), \text{ where } \mathcal{U}(\pi) := \mathbb{E}(U_\pi) = \frac{1}{T} \sum_{t=1}^{T} U_\pi(s_t) \tag{2}$$

**Hindsight Constraint**. We define the scoring function, i.e., pacing function in curriculum learning (Bengio et al., 2009) with $\phi : \mathcal{S} \to \mathcal{G} \times \mathcal{G}$ which is a known and tractable mapping. $\phi$ effectively provides a Bayesian prior $g \in \mathcal{G}$ for data sampling, namely, exploration, where $g$ denotes goal and $\mathcal{G}$ denotes goal space. Based on the analysis above, we can formulate Eq. (1) as

$$\mathcal{U}_g(\pi) = \mathbb{E}_g[U_\pi] = \frac{1}{T} \sum_{t=1}^{T} U_\pi(s_t) \cdot \phi(\cdot|s_t), \tag{3}$$

where $\phi(\cdot|s_t)$ denotes the induced prior probability conditioned on $s_t$. In order to guarantee the convergence, $\phi(\cdot|s_t)$ should always be a non-increasing function of the difficulty level of $s_t$. In our algorithm, we define the goal space $\mathcal{G}$ as a set of visited states in state space $\mathcal{S}$ (i.e., hindsight constraint in Section 3), which guarantees each goal/anti-goal is sampled from previous states. This proves the following result:

**Proposition 1.** The difference between the expected utility function with and without prior $g$ (*i.e.*, $\mathcal{U}_g(\pi)$ and $\mathcal{U}(\pi)$) is the covariance between utility function $U_\pi(s)$ and goal generation $\phi(\cdot|s)$.

*Proof.* The proof of Proposition 1 can be found in Appendix C.3. □

**Diversity Constraint**. However, one should be noted that goal $g$ here is sampled from previous states, which guarantees the reachability of the goal but also limits potential exploration. To address this issue, we adopt diversity measure $\mathcal{H}_{\text{region}}(\pi)$ to encourage the exploration between different region (diversity constraint in Section 3 is a simple implementation). Combining the aforementioned hindsight and diversity constraints, we define our objective as

$$\pi^* = \arg\max_\pi \mathcal{U}_g(\pi), \text{ under hindsight and diversity constraints.} \tag{4}$$

which can be easily derive as equivalence to Eq. (6).

**Proposition 2.** The modified optimization landscape induced by curriculum learning has the same global optimum $\pi^*$ as the original problem.

*Proof.* The proof of Proposition 2 can be found in Appendix C.4. □

According to the analysis, we can conclude that our algorithm can be regarded as a novel curriculum learning approach in a goal-oriented setting, which can be proved to have the same global optimum as the original problem. In Section 4, we conduct experiments to prove that the goals are generated at different levels as the curriculum to guide the agent in curriculum learning.

### B.3 Relationship with Maximum Entropy RL

In this section, we consider multi-goal RL as goal-oriented policy learning (Schaul et al., 2015; Plappert et al., 2018). We further discuss the motivation behind these two constraints, namely hindsight and diversity constraints, and the relationships between our work and inverse maximum entropy reinforcement learning.

**Preliminaries**. We begin with some notations and previous motivations in maximum entropy reinforcement learning (Eysenbach et al., 2020). The likelihood of a trajectory $\tau := \{s_t\}_{t=1}^T$ under policy $\pi$ can be formulated as $\mathcal{L}(s) = \mathcal{P}(s_0) \cdot \prod_t \mathcal{P}(s_{t+1}|s_t, a_t)\pi(a_t|s_t)$. In the goal-oriented RL, we can re-write it as

$$\mathcal{L}(s, g) = \mathcal{P}(s_0) \cdot \prod_t \mathcal{P}(s_{t+1}|s_t, a_t)\pi(a_t|s_t, g), \tag{5}$$

where the initial state is sampled as $s_0 \sim \mathcal{P}(s_0)$ and subsequent states are governed by a dynamic distribution $s_{t+1} \sim \mathcal{P}(s_{t+1}|s_t, a_t)$. As we discuss in Appendix B.2, goal-oriented RL can be regarded as regular RL with prior knowledge $g$ generated by mapping function $\phi$ based $s$. Hence, the target joint distribution over goals and states is

$$\mathcal{L}_{\text{target}}(s, g) = \frac{\phi(\cdot|s)}{Z(g)} \cdot \mathcal{P}(s_0) \prod_t \mathcal{P}(s_{t+1}|s_t, a_t)e^{r(s_t, g_t, \bar{g}_t)}. \tag{6}$$

where $\mathcal{L}_{\text{target}}(s, g)$ be the joint distribution over state $s \in \mathcal{S}$, goal $g \in \mathcal{G}$; and $Z(g)$ is the factor of normalization.

**Diversity Constraint.** We can express the multi-goals RL objective as the reverse KL divergence between the joint state-goal distributions:

$$\max_\pi -\mathcal{H}(s, g) = \max_\pi -\mathcal{D}_{\text{KL}}(\mathcal{L}(s, g)\|\mathcal{L}_{\text{target}}(s, g)) \tag{7}$$

where the joint distribution of likelihood $\mathcal{L}$ and prior information $g$ of a trajectory $\tau$ is defined as $\mathcal{L}(s, g) := \mathcal{L}(s|g) \cdot \phi(\cdot|s)$. Then, we can rewrite Eq. (7) as maximizing the expected (entropy-regularized) reward of a goal-conditioned policy $\mathcal{L}(s|g)$:

$$\mathbb{E}_{g\sim\phi(\cdot|s),\ s\sim\mathcal{L}(\cdot|g)} \left[ \left( \sum_{t=1}^T r(s_t, a_t|g) - \log\pi(s_t, a_t|g) \right) - \log Z(g) \right]. \tag{8}$$

**Hindsight Constraint.** Since the distribution over goals $g$ is fixed, we can ignore the $\log Z(g)$ term for optimization. A less common but more intriguing choice is to factor $\mathcal{L}(s, g) = \phi(\cdot|s) \cdot \mathcal{B}(s)$, where $\mathcal{B}(\tau)$ is represented non-parametrically as a distribution over previously-observed states. Therefore, $\phi(\cdot|s)$ is formulated as a hindsight relabeling distribution. In this implementation, we sample goals from previous states in the region-based memory to present $\mathcal{B}(s)$.

## C Proofs

### C.1 Proof of Proposition 1

**Proposition 3.** *Given the joint set $\mathcal{X}$ and several region-based sets (i.e., sub-sets) $\mathcal{X}_n$, where $n = 1, 2, \ldots, N$ and $N$ is the number of regions, we have*

$$\forall \pi, \ \max_{x\in\mathcal{X}} V(x) \geq \max_{x\in\{x_1, x_2\ldots, x_N\}} V(x), \ \text{where } x_n = \arg\max_{x_n\in\mathcal{X}_n} V(x_n). \tag{9}$$

In this section, we provide the proof of Proposition 1. The motivation of Proposition 1 is to find a relaxed lower bound of $V^{(}x)$, $x \in \mathcal{X}$ based on the definition of the region.

*Proof.* By Eq. (3), $\forall \pi$ we have

$$
\begin{aligned}
\max_{x \in \mathcal{X}} V(x) = \max_{x \in \mathcal{X}} \mathbb{E}_{s \in \mathcal{S}; g, \bar{g} \in \mathcal{G}, \mathcal{P}} & \left[ \sum_{t=1}^{T} \gamma^t r(s_t, a_t | g, \bar{g}) \right] \\
\geq \max_{x \in \{x_1, x_2, \dots, x_N\}} & \Big\{ \max_{x_1 \in \mathcal{X}_1} \mathbb{E}_{s \in \mathcal{S}_1; g, \bar{g} \in \mathcal{G}_1, \mathcal{P}} \left[ \sum_{t=1}^{T} \gamma^t r(s_t, a_t, s_{t+1}) \right], \dots, \\
\dots, \max_{x_N \in \mathcal{X}_N} & \mathbb{E}_{s \in \mathcal{S}_N; g, \bar{g} \in \mathcal{G}_N, \mathcal{P}} \left[ \sum_{t=1}^{T} \gamma^t r(s_t, a_t, s_{t+1}) \right] \Big\} \\
\geq \max_{x \in \{x_1, x_2, \dots, x_N\}} & V^\pi(x), \text{ where } x_i = \arg\max_{x_i \in \mathcal{X}_i} V(x_n), \ n = 1, 2, 3, \dots, N.
\end{aligned}
\tag{10}
$$

The intuition behind the proposition is easy to understand. Since we have partitioned the joint set $\mathcal{X}$ into several region-based sets (i.e., sub-sets) $\{\mathcal{X}_n\}_{n=1}^{N}$. We effectively avoid the agent switching among regions, meanwhile removing these trajectories out of the original candidate trajectory family. $\square$

### C.2  PROOF OF PROPOSITION 4

**Proposition 4.** *Denote the Bellman backup operator in Q learning with goal as $\mathcal{B} : \mathbb{R}^{|S| \times |A| \times |G|} \to \mathbb{R}^{|S| \times |A| \times |G|}$ and a mapping $Q : S \times A \times G \to \mathbb{R}^{|S| \times |A| \times |G|}$ with $|S| < \infty$ and $|A| < \infty$. Repeated applications of the operator $\mathcal{B}$ for our goal-oriented state-action value estimate $\hat{Q}$ converges to a unique optimal value $\hat{Q}^*$.*

*Proof.* The proof of Proposition 4 is done in two main steps. The first step is to show that our goal $g \in \mathcal{G}$ can converge to the terminal state. In the second step, we prove that given goal $g$, our goal-oriented approach can converge to a unique optimal value $Q^*$. In other words, we need to prove that $g \to g^*$ in the first step and $Q \to Q^*$ in the second step.

**Step I.** Our algorithm aims to find the high-value previous states for goal generation. At the beginning of the task, the terminal state will be regarded as the final goal since it has the highest value. Hence, the terminal state, if it has been visited once, will be assigned as the goal. Assume that the agent can conduct plenty of exploration. Then, we can say that the generated goal $g$ will keep approaching the terminal state $g^*$.

**Step II.** Note that the proof of convergence for our goal-oriented RL is quite similar to $Q$-learning (Bellman, 1966; Bertsekas et al., 1995; Sutton & Barto, 2018). The differences between our approach and $Q$-learning are that $Q$-value $Q(s, a, g, \bar{g})$ is also conditioned on goal $g$ and anti-goal $\bar{g}$. As introduced in Section 3, anti-goal $\bar{g}$ works like a reward shaping technique, which is proposed to avoid local optima (Trott et al., 2019). Hence, we omit $\bar{g}$ in the following proof. We provide detailed proof as follows:

We can obtain goal $g \in G$ approaching the terminal state from Step I. Based on that, our estimated goal-conditioned action-value function $\hat{Q}$ can be defined as

$$
\mathcal{B}\hat{Q}(s, a, g) = R(s, a, g) + \gamma \cdot \max_{a' \in A} \sum_{s' \in S} P(s' | s, a) \cdot \hat{Q}(s', a', g).
\tag{11}
$$

For any action-value function estimates $\hat{Q}^1, \hat{Q}^2$, we study that

$$
\begin{aligned}
& |\mathcal{B}\hat{Q}^1(s, a, g) - \mathcal{B}\hat{Q}^2(s, a, g)| \\
& = \gamma \cdot | \max_{a' \in A} \sum_{s' \in S} P(s'|s, a) \cdot \hat{Q}^1(s', a', g) - \max_{a' \in A} \sum_{s' \in S} P(s'|s, a) \cdot \hat{Q}^2(s', a', g)| \\
& \leq \gamma \cdot \max_{a' \in A} | \sum_{s' \in S} P(s'|s, a) \cdot \hat{Q}^1(s', a', g) - \sum_{s' \in S} P(s'|s, a) \cdot \hat{Q}^2(s', a', g)| \\
& = \gamma \cdot \max_{a' \in A} \sum_{s' \in S} P(s'|s, a) \cdot |\hat{Q}^1(s', a', g) - \hat{Q}^2(s', a', g)| \\
& \leq \gamma \cdot \max_{s \in S, a \in A} |\hat{Q}^1(s, a, g) - \hat{Q}^2(s, a, g)|
\end{aligned}
\tag{12}
$$

Combining Step I and II, we can conclude that our goal-conditioned estimated state-action value $\hat{Q}$ can converge to a unique optimal value $Q^*$ leading to the terminal state $g^*$. $\qquad\square$

## C.3 Proof of Proposition 1

In this section, we provide proof of Proposition 1. From Eq. (3), $\mathcal{U}_g(\pi)$ is a function of $\pi$ which is determined by the correlation between $U_\pi(s)$ and $\phi(g)$ (*i.e.*, $\phi(\cdot|s)$). We can rewrite Eq. (3) as

$$\begin{aligned} \mathcal{U}_g(\pi) &= \frac{1}{T}\{\sum_{t=1}^{T}(U_\pi(s_t) - \mathbb{E}[U_\pi])(\phi(g_t) - \mathbb{E}[\phi]) + T \cdot \mathbb{E}[U_\pi]\mathbb{E}[\phi]\} \\ &= \frac{1}{T}\{\mathrm{Cov}[U_\pi, \phi] + T \cdot \mathbb{E}[U_\pi]\mathbb{E}[\phi]\} \\ &= \frac{1}{T}\{\mathcal{U}(\pi) + \mathrm{Cov}[U_\pi, \phi]\} \end{aligned} \tag{13}$$

This derivation can be found in Appendix C.6. We can find that curriculum learning changes the landscape of the optimization function over the policy $\pi$ from $\mathcal{U}(\pi)$ to $\mathcal{U}_g(\pi)$. Intuitively, the above equation also suggests that if the induced goal $g$, which defines a latent variable over the goal space $\mathcal{G}$, is positively correlated with the optimal utility $U_{\pi^*}(s)$, and more so than with any other $U_\pi(s)$, then the gradients in the direction of the optimal policy $\pi$ in the new optimization landscape may be overall steeper.

Hence, this is necessary to design task-related goals. However, it is infeasible to obtain appropriate goals through handcrafted design and manual generation. In this paper, we introduce hindsight and diversity constraints to help the agent learn from achieved task-related information (previous states) and unknown task-related information (unexplored states) respectively.

## C.4 Proof of Proposition 2

In this section, we provide proof of Proposition 2. In order to prove that the modified optimization function in the state-space-related parameter space $\pi$ has the property that the global maximum at $\pi^*$ is more pronounced, we derive the objective function based on Proposition 1. We can assume that optimal policy $\pi^*$ maximizes the covariance between $\phi(g)$ (*i.e.*, $\phi(\cdot|s)$) and utility $U_\pi(s)$, namely

$$\arg\max_\pi \mathcal{U}(\pi) = \arg\max_\pi \mathrm{Cov}[U_\pi, \phi] = \pi^* \tag{14}$$

The proof of the assumption can be found in Appendix C.3. We introduce Lemma 1 here, the proof of which can be found in Appendix C.5.

**Lemma 1.** (Florensa et al. (2017)) For any curriculum satisfying Eq. (14):

1. $\pi^* = \arg\max_\pi \mathcal{U}(\pi) = \arg\max_\pi \mathcal{U}(\pi^*)$
2. $\mathcal{U}_g(\pi^*) - \mathcal{U}_g(\pi) \geq \mathcal{U}(\pi^*) - \mathcal{U}(\pi), \ \ \forall \pi$

Lemma 1 has proposed two claims. The first one presents that the problem of maximizing the covariance between $\phi(g)$ and utility $U_\pi(s)$ shares the same optimal solution with the original problem. In addition, the modified optimization function in the original parameter space without goal $g$ has the property that the global maximum with goal $g$ is more pronounced.

## C.5 Proof of Lemma 1

In this section, we provide the proof of Lemma 1. Claim 1 in Lemma 1 can be derived directly from Eq. (14), while for the claim 2, we have

*Proof.*

$$\begin{aligned} \mathcal{U}_g(\pi^*) - \mathcal{U}_g(\pi) &= \mathcal{U}_g(\pi^*) - \mathcal{U}(\pi) - \mathrm{Cov}[U_\pi, g] \\ &\geq \mathcal{U}_g(\pi^*) - \mathcal{U}(\pi) - \mathrm{Cov}[\mathcal{U}_{\pi^*}, g] \\ &= \mathcal{U}(\pi^*) - \mathcal{U}(\pi) \end{aligned} \tag{15}$$

$\qquad\square$

## C.6 Detailed Derivation of Eq. (13)

In this section, we provide the detailed derivation of Eq. (13). We begin from the formulation of $\mathcal{U}_g(\pi)$ in Eq. (13) and try to obtain that in Eq. (3).

*Proof.*

$$
\begin{aligned}
\mathcal{U}_g(\pi) &= \frac{1}{T}\{\sum_{t=1}^{T}(U_\pi(s_t) - \mathbb{E}[U_\pi])(\phi(g_i) - \mathbb{E}[\phi]) + T \cdot \mathbb{E}[U_\pi]\mathbb{E}[\phi]\} \\
&= \frac{1}{T}\{\sum_{t=1}^{T}(U_\pi(s_t)\phi(g_t)) - \sum_{t=1}^{T}(U_\pi(s_t)\mathbb{E}[\phi]) - \sum_{t=1}^{T}(\phi(g_t)\mathbb{E}[U_\pi]) + T \cdot \mathbb{E}[U_\pi]\mathbb{E}[\phi] + T \cdot \mathbb{E}[U_\pi]\mathbb{E}[\phi]\} \\
&= \frac{1}{T}\{\sum_{t=1}^{T}(U_\pi(s_t)\phi(g_t)) - T \cdot \mathbb{E}[U_\pi]\mathbb{E}[\phi] - \sum_{t=1}^{T}(\phi(g_t)\mathbb{E}[U_\pi]) + T \cdot \mathbb{E}[U_\pi]\mathbb{E}[\phi] + T \cdot \mathbb{E}[U_\pi]\mathbb{E}[\phi]\} \\
&= \frac{1}{T}\sum_{t=1}^{T}(U_\pi(s_t)\phi(g_t)) + \frac{1}{T}\{T \cdot \mathbb{E}[U_\pi]\mathbb{E}[\phi] - \sum_{t=1}^{T}(\phi(g_t)) \cdot \mathbb{E}[U_\pi]\}
\end{aligned}
\tag{16}
$$

Since $\mathbb{E}[\phi] := \frac{1}{T}\sum_{t=1}^{T}(\phi(g_t))$, we have

$$
\begin{aligned}
\mathcal{U}_g(\pi) &= \frac{1}{T}\sum_{t=1}^{T}(U_\pi(s_t)\phi(g_t)) + \frac{1}{T}\{T \cdot \mathbb{E}[U_\pi]\mathbb{E}[\phi] - T \cdot \mathbb{E}[U_\pi]\mathbb{E}[\phi]\} \\
&= \frac{1}{T}\sum_{t=1}^{T}U_\pi(s_t)\phi(g_t)
\end{aligned}
\tag{17}
$$

$\square$

# D  Experiment

## D.1  Modified Environments

**Ant Locomotion**. In this part, we introduce two environments based on Ant Locomotion, namely Free Ant and Ant Maze. The ant is a quadruped with 8 actuated joint, 2 for each leg. The environment is implemented in Mujoco. Besides the coordinates of the center of mass, the joint angles and joint velocities are also contained in the observation of the agent. Considering the high degrees of freedom, navigation in this quite complex task requires motor coordination. More details can be found in Duan et al. (2016), and the only difference is that in our goal-oriented version of Ant, we extend the observation with the goals. The reward is still a sparse indicator function being 1 only when the center of mass $(x, y)$ of the Ant is within $\epsilon = 0.5$ positions corresponding to $\epsilon$-balls in state space. For the Free Ant experiments, the objective is to reach any position in the square $[-5, 5]^2$. Therefore the goal space is 2 dimensional, the state-space is 41 dimensional, and the action space is 8 dimensional. As for the Ant Maze environment, the agent is constrained to move within the maze environment, U-maze in this case, and the size of all the blocks in the maze is $8 \times 8$. The maze consists of a totally 18 blocks.

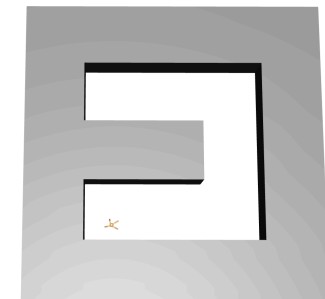

Figure 8: An illustration for Maze Ant Locomotion environment.

**Multi-Path Point Maze**. All the experiment setting is similar to the Ant Maze environment. We replace the Ant agent with a Point-Mass and change the maze into a multi-path one. The action of the Point-Mass is a velocity vector, namely, in the 2 dimension.

**$N$-dimensional Point-Mass Maze**. In the $N$-dimensional Point-Mass maze experiment, the agent can only move within a small subset of the state space. In the two-dimensional case, the set of feasible states corresponds to the $[-5, 5] \times [-1, 1]$ rectangle, making up 20% of the full space. For $N > 2$, the feasible space is the Cartesian product of this 2D strip with $[-\epsilon, \epsilon]^{N-2}$, where $\epsilon = 0.3$.

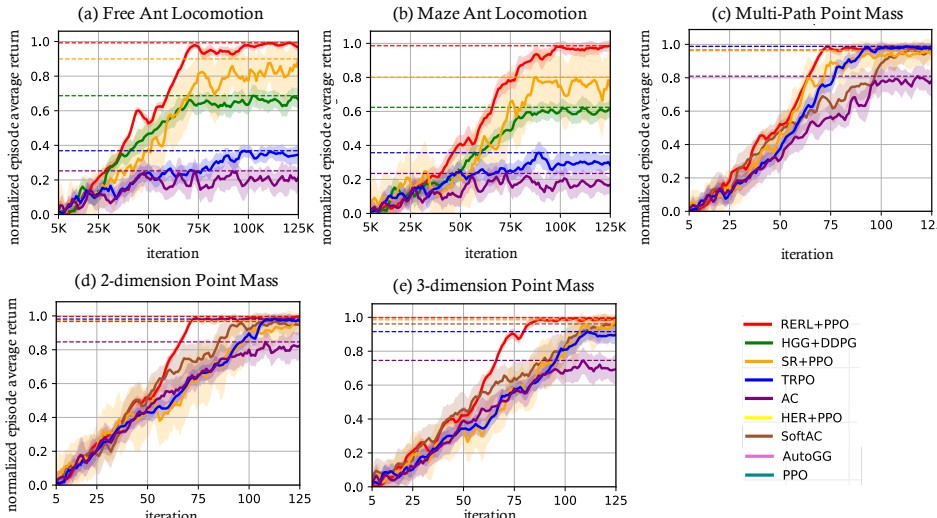

Figure 9: Learning curves of RERL, HGG, HER, SR and AutoGG on various environments, where the solid curves depict the mean, the shaded areas indicate the standard deviation, and dashed horizontal lines show the asymptotic performance.

In this higher-dimensional environment, our agent receives a reward of 1 when it moves within $\epsilon_N = 0.3 \frac{\sqrt{N}}{\sqrt{2}}$ of the goal state, to account for the increase in average $L2$ distance between points in higher dimensions. In these experiments, the full state-space of the $N$-dimensional Point Mass is the hypercube $[-5, 5]^N$.

## D.2 EVALUATION DETAILS

We adopt HGG (Ren et al., 2019) incorporating with DDPG (Lillicrap et al., 2015), SR (Trott et al., 2019) accompanying with PPO (Schulman et al., 2017) as these models are originally proposed. All curves presented in this paper are plotted from 12 runs with random task initializations and seeds. Following the regular procedure in goal-oriented RL, an episode is considered successful if and only if the agent obtain 1 as the reward according to Eq. (2) where $\delta$ stays the same for all the approaches. However, in the practice, we conduct reward as the $r(s_t, a_t | g, \bar{g}) = \min[0, -d(\phi(g_{t+1}|s_{t+1}), g) + d(\phi(\bar{g}_{t+1}|s_{t+1}), \bar{g})]$ to accelerate the training process.

## D.3 IMPLEMENTATION DETAILS

Almost all hyper-parameters using DDPG (Lillicrap et al., 2015), TRPO (Schulman et al., 2015), PPO (Schulman et al., 2017), Soft-AC (Haarnoja et al., 2018) are kept the same as benchmark results. Specifically, we list our hyper-parameters as here. number of MPI workers: 1; buffer size: $10^4$ trajectories; number of regions $N$: 5 in agent level; batch size: 256, number of trajectories $Z$: 50, Lipschitz constant $L$: 5; learning rate: $10^{-5}$ in the network level; discount factor: 0.99; interpolation factor in Polyak averaging (if there is): 0.995; scale of additive Gaussian noise: 0.2; probability of HER (Andrychowicz et al., 2017) experience replay: 0.8.

# E RESULTS

## E.1 ADDITIONAL EVALUATION ON STANDARD TASKS

In this section, we provide additional results on comparisons between RERL and various baselines.

In order to answer the first two questions, we demonstrate our method in two challenging robotic locomotion tasks, where the goals are the $(x, y)$ position of the center of mass of a dynamically complex quadruped agent. In the first example, the agent has no constraints, and in the second one, the agent is inside a U-maze (see Section 4 for details). Results in Figure 9(a)(b) demonstrate that the performance of our approach exceeds that of the strong baselines mentioned in Section 4. To answer the third question, we train an ant agent to reach any position within a multi-path maze. As

shown in Figure 9(c), our approach obtains better performance even in the multi-path environment where goal distribution is naturally more complex than previous environments. To answer the fourth question, we investigate how our method performs with the dimension of goal-space in an environment where the goal space grows in dimension within the feasible region, *e.g.*, 2D and 3D. As shown in Figure 9(d), our approach outperforms strong baselines in both low- and high-dimensional environments.

### E.2 ADDITIONAL RESULTS ON VISUALIZATION OF GENERATED GOALS

To answer the final question, we conduct a visualization study on generated goals to investigate whether goals can encourage the agent to the target state, and anti-goals can prevent the agent from the local optima. The visualization of goals can also represent the effect of diversity and hindsight constraints through exploration and reachability of generated goals.

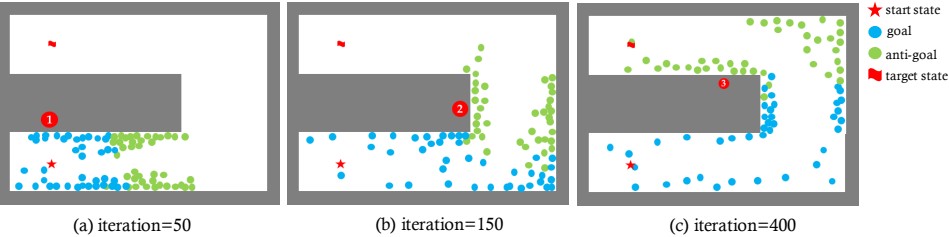

Figure 10: Generated goals and anti-goals visualized as the blue and green points respectively.

Results in Figure 10 show that the hindsight constraint helps the agent aim at feasible positions while our diversity constraint encourages the agent to approach the target state. Specifically, from ① and ②, one can note that the agent is pulled by its goal and pushed by its anti-goal and goals from the other regions. Hence, once a region is leading to a wrong direction, it also can encourage exploration via diversity constraint.

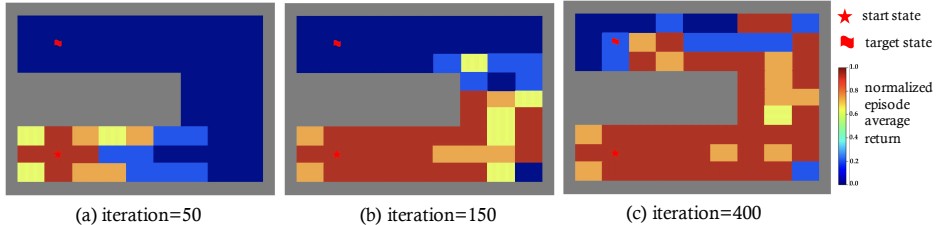

Figure 11: Each grid cell in U-maze is colored according to the expected return success rate when fixing its center as the target state.

As illustrated in Figure 11, the generated goals are approaching as the training proceeds, and at an appropriate success rate level, which is accorded with the curriculum in the curriculum learning (see Appendix B.2 for details).

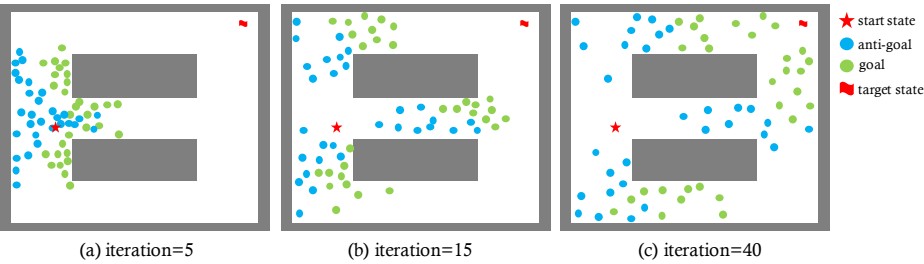

Figure 12: Generated goals and anti-goals visualized as the blue and green points respectively.

Results showed in Figure 13 and 12 are similar with that in Figure 11 and 10 respectively, which actually can confirm the analysis above.

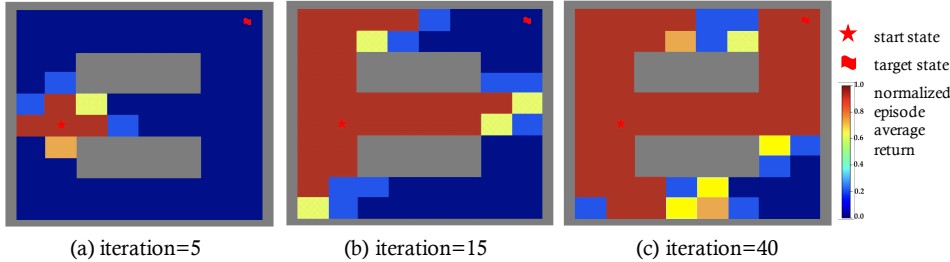

(a) iteration=5        (b) iteration=15        (c) iteration=40

Figure 13: Each grid cell in Multi-path maze is colored according to the expected return success rate when fixing its center as the target state.

### E.3 EXPERIMENT ON THE COMPARISON WITH EXPLICIT CURRICULUM LEARNING

In (Florensa et al., 2017), GOID is defined as a goal set as $\mathrm{GOID}(\pi) = \{g : \alpha \leq f(\pi, g) \leq 1 - \alpha\}$ where $f(\pi, g)$ represents the average success rate in a small region closed by goal $g$. In order to construct the GOID set, we follow its definition and sample generated goals from $\mathrm{GOID}(\pi)$ via rejection sampling.

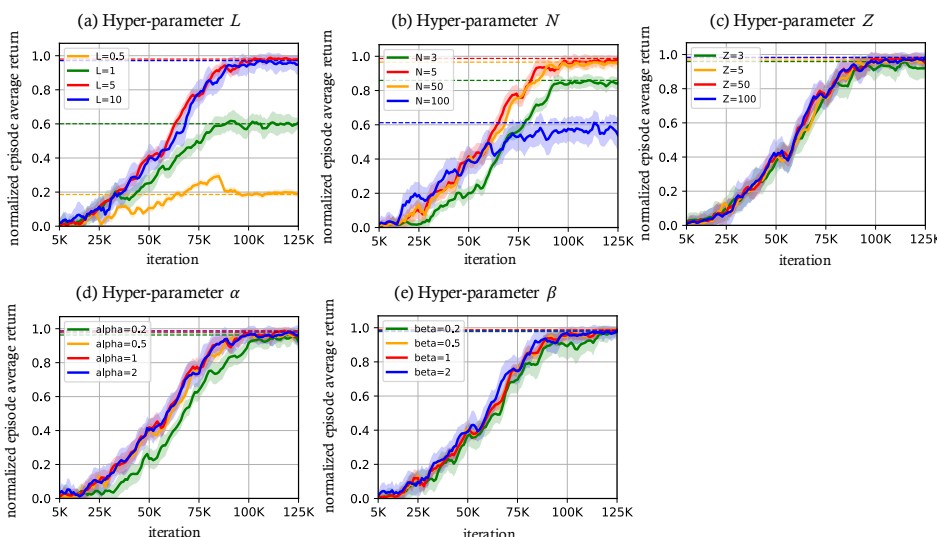

Figure 14: Learning curves of ablation study on parameters: $L$, $N$, $Z$, $\alpha$ and $\beta$, where the solid curves depict the mean, the shaded areas indicate the standard deviation, and dashed horizontal lines show the asymptotic performance.

### E.4 ADDITIONAL RESULTS ON ABLATION STUDY

In this section, we set up a set of ablation tests on several hyper-parameters used in the RERL. The selection of Lipschitz constant $L$ is task-dependent since it is highly related to the scale of the value function and goal distance. For the robotics tasks tested in this paper (i.e., Ant Maze Locomotion), as showed in Figure 14(a), we find that the performance of RERL is reasonable as long as $L$ is not too small. Similar to $L$, the selection of the number of regions $N$ is also theoretically task-specific. We test a few choices on Ant Maze Locomotion and find a range of $N$ that works well. As Figure 14(b) illustrates, it appears that the RERL is reasonable as long as $N$ is not too large. As for the number of trajectories $Z$, we plot the curve on different $Z$ in Figure 14(c) and find that for the simple tasks, the choice of $Z$ is not critical. Parameters $\alpha$ and $\beta$ together define the trade-off between value function, diversity, and hindsight constraints. Results in Figure 14(d)(e) show that the choice of $\alpha$ and $\beta$ is indeed robust.

