# OpenReview forum: "Regioned Episodic Reinforcement Learning"
_ICLR.cc/2021/Conference — Reject_

### Official Review · AnonReviewer3 · 2020-10-26
**Combining goal oriented RL with episodic memory**

**Rating:** 5
**Confidence:** 3

**Review:**

**Summary:**
This paper presents a new algorithm called Regioned Episodic Reinforcement Learning (RERL), which combines ideas from episodic memory, with automatic sub-goal creation or “goal-oriented” RL. The method works by dividing the state space into regions, where a different goal identifies each region. Then, using an episodic memory technique, the agent is able to learn about new experiences in a sample efficient way. This allows the agent to explore effectively, and learn a good policy quickly in problems where there are sparse rewards. The paper provides some theoretical justification for the new algorithm, and provides some empirical results that demonstrate its effectiveness.

**Strengths:**
The paper combines two RL techniques in a novel way to address important issues in deep rl. The experiments demonstrate that RERL can achieve good performance on challenging problems. The paper also presents a good background overview of related work, which is useful to get appropriate context.

**Weaknesses:**
The use of the term episodic is a bit ambiguous, especially in the introduction. The word “episodic”, as it refers to “episodic memory” from neuroscience, needs to be clearly disambiguated the from the “episodic setting” in RL, which is something completely different. Additional clarification in abstract/introduction would do a lot to reduce confusion.

The paper has many experiments on a variety of domains, but only a few of the results are included in the main text. I think including more of the results in the main paper would strengthen the conclusions and be more convincing for the reader. Also, some important experiment details are missing, like how many training steps was each agent trained for (what do "iterations" correspond to in Figure 5)? The number of independent runs (seeds) for each algorithm should also be listed. This information should be included so that the reader can better evaluate the variance of each method, and make more confident conclusions.

**Recommendation:**
I vote to accept this paper. I think this paper presents a novel and interesting idea that is sufficiently supported by the empirical experiments. The challenges that the new methods attempts to address and important and central to good performance in deep RL, thus the papers results would be of interest to RL experimentalists and practitioners. However, I think the paper could be made stronger improving the clarity, and by including more of the experiments in the main text.

**Questions:**
How many independent runs (seeds) were used in the experiments?

It was not clear to me whether the anti-goal $\bar g$ is new in this paper, or if it is something that has been used before.

More about $\bar g$: the paper states that it “prevents the policy from getting stuck at the local optimum and enables the agent to learn to reach the goal location quickly,” but does not offer an explanation why. Why does it keep the agent from getting stuck at a local optimum? And why does it help the agent reach the goal location quickly?

**Minor Comments:**
-I find the hyperlink highlighting in the text (the green and red boxes) to very distracting. I think removing them would improve the overall presentation of the paper.
-The in text citation style could be improved. When citing an author and using their name in the text, they should be cited like “Foo et al. (2020) propose a new method…”, and not like “(Foo et al., 2020) propose a new method…”
(For more details see: https://owl.purdue.edu/owl/research_and_citation/apa_style/apa_formatting_and_style_guide/in_text_citations_the_basics.html, for example.)
-There is a missing figure reference on page 7.

**After Author Response and Discussion:** Thanks to the authors for their responses. After reading the other reviews and the author responses, I am lowering my score to 5. I agree with the other reviewers about the experiments: they feel rushed, and the design and presentation is not as careful as it could be. I think the paper could be greatly improved by a little more attention the presentation of the experiments and their analysis in the main text.

---

> ### Author Response · Authors · 2020-11-20
> **Response to Review #3**
>
> Thank you for your feedback and suggested experiments. Please also see the main comment above. We hope you will consider increasing your score after seeing our responses.
>
> >> The use of the term episodic is a bit ambiguous, especially in the introduction. The word “episodic”, as it refers to “episodic memory” from neuroscience, needs to be clearly disambiguated the from the “episodic setting” in RL, which is something completely different. Additional clarification in abstract/introduction would do a lot to reduce confusion.
>
> Thanks for your suggestion. We have clarified this in the revision. The “episodic” in this paper refers to “episodic memory” since RERL constructs and updates region-based memories to perform efficient updates. The “episodic setting” in RL may mean the environment is under an episodic setting, which is different from ours.
>
> >> I suggest including more of the results in the main paper. Also, some important experiment details are missing, like how many training steps was each agent trained for (what do "iterations" correspond to in Figure 5)?
>
> Great suggestion. We have increased the number of results in the main paper to 10. The "iterations" in Figure 5 correspond to total timesteps.
>
> >> How many independent runs (seeds) were used in the experiments?
>
> 12 independent seeds were used in the experiments.
>
> >> It was not clear to me whether the anti-goal g¯ is new in this paper, or if it is something that has been used before. Why does it keep the agent from getting stuck at a local optimum? And why does it help the agent reach the goal location quickly?
>
> The anti-goal g¯ is not new in this paper. It is first proposed in [1]. However, different from [1] where anti-goal g¯ is generated by rollouts, we set the anti-goal as the state with the average value. The reason behind it is that it is hard to judge whether the agent performs well in each episode since the agent is learning and its performance is getting better. Therefore, we propose to use anti-goal as a baseline for it. Figure 2 in [1] illustrates that the anti-goal setting can avoid the agent from getting stuck in a local optimum.
>
> [1] Alexander Trott, et al. Keeping Your Distance: Solving Sparse Reward Tasks Using Self-Balancing Shaped Rewards. NeurIPS, 2019.

---

### Official Review · AnonReviewer4 · 2020-10-29
**The method section is not easy to follow. Results deserve more attention and clearer discussion**

**Rating:** 5
**Confidence:** 4

**Review:**

### Summary
This work proposes a curriculum learning in RL that combines goal-oriented RL and episodic RL in order to achieve a sample efficient exploration-exploitation trade-off.

Clarity: The paper could benefit from a clear presentation in order to help properly understand the proposed methods and all its components.

### Contributions
- The paper designs an intrinsic reward to encourage exploration while avoiding local optima policies.
- Exploitation is leveraged through a region-based memory.
- The paper proposes a goal sampling scheme that combines hindsight constraint for reachability and diversity constraint for exploratory behaviour.

### Concerns
- \mathcal{X} is sometimes used as a set sometimes as a distribution. This might raise some confusion for the reader.
- The mapping \phi is quite central to you method, but I failed to understand which type of mapping was used in your experiments. Consequently, which metric was adopted ?
- "we directly assign the visited state with the average value in the region as the anti-goal.": Does this mean, the anti-goal is always in the same region as the visited state ? Can you please clarify this point/choice of anti-goal?
- In eq.9, how does V^\pi(x_s) depend on s_t (is it integrated to the inner minimization)?
- "slowly changing goal-conditioned tasks to guarantee stability by restricting goal updating within each region": Do we need to guarantee slow (spatial) changing of the sampled region too for this stability to hold ? If yes, how is this enforced in the algorithm?
- How were the existing approaches trained ? With which reward ? the one from eq.2 (i.e. using the region-based goal and anti-goal sampling ) ?
- "We further extend the task to evaluate whether the agent is able to reach any given position (\epsilon-balls depicted in red) within the maze for Maze Ant": What was the size of the agent (Ant) compared to the width of the corridor ? *Ant Maze look tight around agent, i.e if it learns to reach the end of the maze it can’t miss any intermediate goal.*
- From the goals and anti-goals visualizations, anti-goals seems to be always generated further ahead on the path to the target, while it was mentioned that the anti-goal is set to the average value in the region ? When getting closer to the target, won't these anti-goals between the reachable goals and the target keep the agent away from reaching it ? Can you please clarify this ?
- Can you elaborate on the goal generation ablation study ? This seem to deserve more attention to better evaluate the algorithm.

### Minor comments
- Some notations are defined later in the paper than where they were first introduced.
- In eq.9, should the argument be x_z ?
- There is a problem with some figures and appendices references.
- Did you try higher dimensional Point-Mass ?

---

> ### Author Response · Authors · 2020-11-20
> **Response to Review #4**
>
> Thank you for your feedback and suggested experiments. Please also see the main comment above. We hope you will consider increasing your score after seeing our responses.
>
> >> \mathcal{X} is sometimes used as a set, sometimes as a distribution. This might raise some confusion for the reader.
>
> Thanks for pointing it out. In our method, \mathcal{X}_{n} is a set and we have revised our paper to avoid such confusion.
>
> >> The mapping \phi is quite central to your method, but I failed to understand which type of mapping was used in your experiments. Consequently, which metric was adopted?
>
> \phi is a mapping function from state space to goal space which largely depends on an environment. For example, in the Ant Maze case, this mapping is to map the system state, which is a high-dimensional vector including information about the robot and the environment, to the 3-D position of the robot Ant. The mapping function \phi is usually provided by an environment, such as gym robot environments.
>
> >> "we directly assign the visited state with the average value in the region as the anti-goal.": Does this mean, the anti-goal is always in the same region as the visited state? Can you please clarify this point/choice of anti-goal?
>
> Yes. The reason behind it is that it is hard to judge whether the agent performs well enough in each episode and in each region, since the agent is learning and its performance is getting better. Therefore, we propose to use anti-goal as a baseline for it.
>
> >> In eq.9, how does V^\pi(x_s) depend on s_t (is it integrated to the inner minimization)?
>
> Yes. Since x_s = (s, g, g¯), we directly select x_s where s (i.e., s_t) can obtain the minimization.
>
> >> "slowly changing goal-conditioned tasks to guarantee stability by restricting goal updating within each region": Do we need to guarantee slow (spatial) changing of the sampled region too for this stability to hold? If yes, how is this enforced in the algorithm?
>
> Yes. Our method adopts Boltzmann softmax to select appropriate regions. And the temperature of the softmax mechanism is changing.  At the beginning of training, the selection is nearly random to avoid richer-get-richer phenomena. In the middle stage of training, the Boltzmann softmax focuses on the region containing the state with the highest value. Then, only when the current region is overestimated will the agent switch to another region. Usually, it takes several rounds of updates to figure out whether the current region is overestimated.
>
> >> How were the existing approaches trained? With which reward? the one from eq.2 (i.e. using the region-based goal and anti-goal sampling)?
>
> Yes. Existing goal-oriented RL approaches are trained using intrinsic rewards as shown in Eq2.
>
> >> "We further extend the task to evaluate whether the agent is able to reach any given position (\epsilon-balls depicted in red) within the maze for Maze Ant": What was the size of the agent (Ant) compared to the width of the corridor? Ant Maze look tight around agent, i.e if it learns to reach the end of the maze it can’t miss any intermediate goal.
>
> Fig.4 (b) in our paper presents a typical example of the AntMaze environment. In practice, the size of the width of the corridor compared to the agent(Ant) is about 6, which guarantees that reaching any given position is a challenging task. The direct rendering result of the AntMaze environment we use is shown in Fig.8 in Appendix D.
>
> >> From the goals and anti-goals visualizations, anti-goals seems to be always generated further ahead on the path to the target, while it was mentioned that the anti-goal is set to the average value in the region ? When getting closer to the target, won't these anti-goals between the reachable goals and the target keep the agent away from reaching it ?
>
> Yes. The labels of the original figure were reversed. In that figure, anti-goals were labeled as goals and goals as anti-goals. We have revised this in the updated version. In that figure, one can note that the agent is pulled by its goal and pushed by its anti-goal and goals from other regions, which is in line with our intuitive explanation.
>
> >> Can you elaborate on the goal generation ablation study ? This seem to deserve more attention to better evaluate the algorithm.
>
> Yes, great suggestion. We have provided detailed descriptions of the setting in the ablation study (see Section 4. Impact of Goal Generation, and Impact of Hyper-parameter Selection). We have evaluated RERL with various RL algorithms in Figure 5(f), and various environments in Figure 5(a)(b)(c)(d)(i).
>
> >> Minor comments
>
> Thank you for pointing out these problems. We have revised and re-written these parts in the paper. Please refer to the revised PDF. We have tried higher dimensional Point-Mass. The result is consistent with the 3-dimensional one. If you think that this result should also be reported, we will update the result in the appendix in the next revision.

---

> > ### Comment · AnonReviewer4 · 2020-11-24
> > **Thank you for your response**
> >
> > Thank you for the clarifications that helped with some of my confusions. I'll update my review accordingly.

---

### Official Review · AnonReviewer1 · 2020-10-29
**Neat idea, insufficient experimental design/analysis**

**Rating:** 5
**Confidence:** 4

**Review:**

The paper proposes a reinforcement-learning scheme intended to strike a productive balance between exploration and exploitation by way of decomposing the state space into regions and developing policies for "solving" each region.

Although the paper claims to include "extensive experiments", the design and analysis of the method was rather coarse. In particular, the results presented in Figure 5 seem solid. But, based on the presentation up to this point, I was expecting to see a comparison on an environment that is good for goal-oriented methods and one that is good for episodic algorithms. I thought you'd show your approach working well in both, while the other methods struggle when used outside of their ideal setting.

My main frustration was that a lot of the paper was spent presenting the algorithm and then the experiments were presented only very briefly. For an algorithm like this, its utility is measured in its ability to work according to its design. I agree that showing the algorithm outperforming state-of-the-art algorithms on the ant problems is a good thing. But, we wouldn't expect that the algorithm would outperform the others in ALL environments, would we? If not, then seeing some assurance that the reason that it is working well is because the problem structure somehow matches the algorithm structure would be very valuable. Then, we can see it's not just luck but its DESIGN that is the cause of its experimental success. The way the results are currently presented, I'm left with the feeling that the algorithm is clever but perhaps limited in scope. More detailed experiments would help allay this fear.

The paper needs a detailed editing pass. Comments (hopefully helpful!) follow.

Detailed comments:

"by current policy" -> "by the current policy"?

"these methods investigating to explore useful trajectories efficiently" -> "these methods intended to explore useful trajectories efficiently"?

"suffer from generating appropriate goals" -> "suffer from the difficulty of generating appropriate goals"?

C2: Perhaps reword it? I wasn't able to interpret what is being said. I think there is a lot of important background being left unsaid.

"also make substantial contributions to improve" -> "also make substantial contributions that improve"?

"when updating value function" -> "when updating the value function"?

"enforcing goal space to be" -> "enforcing that the goal space be"?

"a subset of state space": As opposed to what? Doesn't the state space include ALL states? So, any set of states is a subset of the state space...?

"exploration issue automatically" -> "exploration automatically"?

"with hindsight constraint" -> "using a hindsight constraint"

"the whole exploration space": What's an exploration space? Is it the entire state space?

"perform the goal-oriented RL on" -> "perform goal-oriented RL on" or "execute the goal-oriented RL algorithm on"?

"utilizes diversity constraint in goal generation procedure" -> "utilizes a diversity constraint in the goal generation procedure"

"in human decision-making procedure" -> "in human decision making"

"investigated to integrate" -> "investigated integrating"

"with deep Q network (DQN)" -> "with deep Q networks (DQNs)"

"have proposed" -> "have been proposed"

"to estimate value function" -> "to estimate value functions"?

"use look-up operation" -> "use look-up operations"

"There are mainly two folds" -> "There are mainly two lines of work"?

"via high-level actions or goal" -> "via high-level actions or goals"

"discovering these sub-tasks or sub-goals, which is easy to reach" -> "discovering sub-tasks or sub-goals that are easy to reach"

"in (Nachum et al., 2018)," -> "by Nachum et al. (2018),"

"constraining the goal generation within a specific region under hindsight constraint." -> "constraining goal generation to within a specific region under a hindsight constraint."?

"RL problem can" -> "The RL problem can"?

"In the setting of finite horizon" -> "In the finite-horizon setting"

"state-value function Q" -> "state-action value function Q"

"tuple from a replay" -> "tuples from a replay"?

"training, which is a typical parametric RL methods, suffering from sample inefficiency" -> "training. It is a typical parametric RL method and suffers from sample inefficiency"

"in the environment with sparse" -> "in environments with sparse"

"compute sparse reward"?

"an agent ... yields sparse rewards"? Maybe I'm not understanding how you are using the phrase "sparse rewards"? I interpret it as an environment where most rewards are zero, so I don't see how they are "computed" or "yielded".

"Specially," -> "Specifically,"?

"should base on" -> "should be based on"?

"of diversity constraint" -> "of the diversity constraint"

"avoid the local optima" -> "avoid local optima"

"can be further explained by" -> "and can be further described as"

"Appendix B.1" -> "Appendix B.1."

"tasks to guarantee stability" -> "tasks guarantees stability"?

"be well adaptive with" -> "adapted to"?

"etc.": Redundant, given "such as"? (Twice.)

"(see Figure ??)"?

"accompanying with" -> "accompanying" or "with" or "accompanied by"

"One should be noted" -> "Note"

"can be proved as" -> "can be shown to be"? Not quite. I'm not sure how to reword it, but some rewording is needed.

In terms of related work, I was hoping to hear how you see your work relating to Konidaris' work on skill discovery. There definitely seems to be some echoes in terms of overall algorithmic strategy.

---

> ### Author Response · Authors · 2020-11-20
> **Response to Review #1**
>
> Thank you for your detailed feedback and suggested experiments. Please also see the main comment above. We hope you will consider increasing your score after seeing our responses.
>
> >> I was expecting to see a comparison on an environment that is good for goal-oriented methods and one that is good for episodic algorithms. I thought you'd show your approach working well in both, while the other methods struggle when used outside of their ideal setting.
>
> Thanks for your suggestion. We have conducted experiments on the Atari game Pong. The episodic algorithms such as [1][2] work well in the environment. We report the result in figure 5(d). Typical goal-oriented environments provide explicit mapping function \phi to map the state space to the goal space. However, Atari games do not provide such function and it will make goal-oriented RL hard to converge. To be specific, in the Pong environment, it’s difficult to define goals since \phi is not defined. Thus the performance of methods involving the goal-settings struggles to achieve good performance.
>
> >> seeing some assurance that the reason that it is working well is because the problem structure somehow matches the algorithm structure would be very valuable. Then, we can see it's not just luck but its DESIGN that is the cause of its experimental success. The way the results are currently presented, I'm left with the feeling that the algorithm is clever but perhaps limited in scope. More detailed experiments would help allay this fear.
>
> We have conducted more experiments on both the environments suitable for goal-oriented RL algorithms (e.g., Maze Ant Locomotion) and the environment suitable for episodic RL algorithms (e.g., Atari Game). Results show that RERL obtains good performance in all goal-oriented environments and bad performance in Atari games. Such a performance gap is due to the difficulty in defining the distance in the state/goal space for Atari games. Experiments show that our design generates promising and explainable results for environments within the scope of Goal-Oriented RL. However, for environments where goal space is inexplicit, our algorithm cannot guarantee performance. We also provide the result of the visualization study in Figure 6, which implies that our goal generation approach can truly generate appropriate goals to encourage exploration.
>
> >> C2: Perhaps reword it? I wasn't able to interpret what is being said. I think there is a lot of important background being left unsaid.
>
> Thanks for your suggestion and we have re-written C2. The idea behind C2 is that training goal-oriented RL models using all historical trajectories rather than selected ones would involve unrelated trajectories in training. The training process of goal generation algorithms thus could be unstable and inefficient, as data distribution shifts when the goal changes. It can be fairly efficient if updates happen only with highly related trajectories.
>
> >> "a subset of state space": As opposed to what? Doesn't the state space include ALL states? So, any set of states is a subset of the state space...?
>
> Yes, it was meant to say that the goal space is a set of visited states. We have clarified it in the paper.
>
> >> "an agent ... yields sparse rewards"? Maybe I'm not understanding how you are using the phrase "sparse rewards"? I interpret it as an environment where most rewards are zero, so I don't see how they are "computed" or "yielded".
>
> Yes, the sparse rewards are generated by the environment and used by an agent.
>
> >> In terms of related work, I was hoping to hear how you see your work relating to Konidaris' work on skill discovery.
>
> We include these references in the related work section and discuss the difference between this work and his work. The common idea we share is to break the original task into several sub-tasks. The main difference is that the sub-tasks are getting harder in the training procedure, while his method aims to find the option that can be used to accelerate the learning procedure.
>
> >> The paper needs a detailed editing pass.
>
> Thank you for pointing out these problems. We have revised and re-written these parts in the paper. Please refer to the revised PDF.
>
> [1] Adrià Puigdomènech Badia et al. Agent57: Outperforming the human Atari benchmark. ICML, 2020.
>
> [2] Zichuan Lin, et al. Episodic Memory Deep Q-Networks. IJCAI, 2018.

---

### Official Review · AnonReviewer2 · 2020-10-30
**Promising approach, but not ready for publication.**

**Rating:** 4
**Confidence:** 4

**Review:**

Authors present an approach to partition the state space by generating a diverse set of goals, and to explore the state space effectively by finding policies that can reach to these goals effectively.

At each step, the algorithm chooses a region (or a goal) to explore by sampling from a Boltzmann soft (arg) max distribution with an annealing temperature parameter.

The goal generation strategy is strengthened by two heuristic ideas, namely adding hindsight and diversity constraints.

Overall, while I think breaking a large RL problem to several tractable sub-problems is an interesting direction, I think that the paper is not doing a good job of justifying the specific way in which this is performed.

At a high-level, the approach could be thought of as finding a set of options (along with finding option goals and initiation sets), and then searching for the kind of behavior that maximizes option value. With this in mind, I expect the authors to better situate their work in comparison with option-discovery approaches in RL such as "Eigenoption discovery through the deep successor representation" Machado et al., "Option Discovery using Deep Skill Chaining" Bagaria and Konidaris, and "Exploration in reinforcement learning with deep covering options" Jinnai et al., to name a few.

Some additional comments and questions:

- To say that "the application of reinforcement learning (RL) is still impractical in terms of sample efficiency" is inaccurate. Sample inefficiency is definitely an issue that can plague RL in some applications, but that does not mean that RL is impractical regardless of the application in question.

- I don't understand what it means to say some RL algorithms "neglect the exploitation part". Surely, goal-oriented RL performs reward maximization in some sense, right?

- The instability of RL is, to the best of my knowledge, not related to sparse rewards (C1). Among other things, it is related to convergence guarantees (or lack thereof) when using nonlinear approximators and highly-correlated updates. Can you provide citations that refute my claim and support yours?

- I don't even understand C2 in introduction.

- with C3, I understand what you are trying to say, but maybe you can rephrase it? Maybe say, breaking a task to a bunch of sub-tasks can avoid redundant exploration?

- wrong citation for DDPG.

- in general I find frequent typos and mistakes. For example, from related work, you say "value propagation methods have proposed ..." Humans propose to use value propagation methods that obtain trajectory-centric values estimates.

- why is the range for the reward signal open? Why can't R be 1?

- You consider domains with continuous actions, which means that the policy outputs a pdf over continuous space. It is then, not accurate to say that the policy output is bounded above by 1.

- In equation (1), the notation for reward is off.

-"However, as stated in (Pritzel et al., 2017), in the environment with sparse rewards, there may be very few instances where the reward is non-zero, making it difficult for an agent to find good past experiences. " Does this incredibly obvious point require a citation?!

- From Definition 1, can you clarify what even is a "perfect" partition? We should be able to define it before attempting to find it.

- In equation 5, how is max computed if the space is continuous?

- In formulation 10, why did you choose this specific way of promoting diversity? It is not justified, and there is no theoretical evidence to demonstrate that this indeed gives us the diversity that you tout it would.

---

> ### Author Response · Authors · 2020-11-20
> **Response to Review #2**
>
> Thank you for your feedback and suggested experiments. Please also see the main comment above. We hope you will consider increasing your score after seeing our responses, updates, and new experiments.
>
> >>  situate their work in comparison with option-discovery approaches in RL
>
> Thanks for your suggestion. We revised the related work section and included all suggested references in the related work section. In addition, we ran new experiments and compared our method with “Exploration in reinforcement learning with deep covering options" (Jinnai et al. 2020) [called OPTION in Figure 5]. Please see Figure 5 in the experiment section.
>
> >> To say that "the application of reinforcement learning (RL) is still impractical in terms of sample efficiency" is inaccurate.
>
> We agree with the reviewer that a sample inefficient RL algorithm can be still practical.  However, an RL method can be very cost-prohibitive for many real-world applications such as robotic, autonomous driving, etc. due to data collection cost (e.g. interaction with an environment). We updated the paper to clarify this.
>
> >> say some RL algorithms "neglect the exploitation part".
>
> You are right. We have updated the paper to better describe this problem and the paragraph is re-written as “These methods intend to explore more unique trajectories and use all trajectories in the training procedure, which may involve unrelated ones and result in inefficient exploitation.”
>
> >> The instability of RL is, to the best of my knowledge, not related to sparse rewards (C1).
>
> You are right. We have updated the paper. Sparse reward issue mainly results in the long training time, while the instability of RL usually means failing in the training.
>
> >> I don't even understand C2 in introduction.
>
> We have re-written C2, as follows: “Training goal-oriented RL models using all historical trajectories would involve unrelated trajectories in training. The training process of goal generation algorithms could be unstable and inefficient (Kumar et al.,2019), as data distribution shifts when the goal changes. It can be fairly efficient if updates happen only with highly related trajectories.”
>
> >> with C3, I understand what you are trying to say, but maybe you can rephrase it?
>
> Thanks for your kind suggestion and we have rephrased it.
>
> >> in general I find frequent typos and mistakes.
>
> Thank you for pointing out these problems. We have revised and re-written these parts in the paper. Please refer to the revised PDF.
>
> >> why is the range for the reward signal open?
>
> That was a typo and  R can be 1. Please refer to the revised and updated PDF.
>
> >> You consider domains with continuous actions, which means that the policy outputs a pdf over continuous space. It is then, not accurate to say that the policy output is bounded above by 1.
>
> It is right that policy outputs a pdf over continuous space. We clip it by 1 if it is above by 1. Then, the policy output is bounded between 0 and 1. It is a standard practice of policy networks. Similar approaches can be found in SAC [1], TD3 [2].
>
> >> Does this obvious point require a citation?
>
> That point is obvious and easy to understand. We have removed that citation.
>
> >> From Definition 1, can you clarify what is a "perfect" partition? We should be able to define it before attempting to find it.
>
> The perfect partition of regions should satisfy the following properties:  (1) all the regions together cover the state space,  (2) there is no overlapping among all the regions,  (3) the goal generated from each region can not lead the agent to other regions. Actually, these three properties are the motivation of our diversity constraint design. Thanks a lot for this suggestion. We have updated the paper.
>
> >> In equation 5, how is max computed if the space is continuous?
>
> We found that the original definition of \mathcal{X}_{n} was confusing and have reformulated this equation. We compute the max for trajectories in a specific regioned memory. For instance, suppose that we have 5 trajectories in the upper region and 5 trajectories in the lower region in the AntMaze environment. The max here is computed for trajectories in each region separately. Such computation can be done if the space is continuous.
>
>
> >> In formulation 10, why did you choose this specific way of promoting diversity?
>
> The motivation of the diversity constraint is to avoid the overlapping among all the regions. Note that trajectories in each region will be guided by a goal. Hence, we design to maximize the distances among the generated goals to automatically construct the regions without overlapping. We have edited the paper to make it more clear and the experiments show that this constraint can prevent the regions from overlapping.
>
> [1] Tuomas Haarnoja, et al. Soft Actor-Critic: Off-Policy Maximum Entropy Deep Reinforcement Learning with a Stochastic Actor. ICML, 2018.
>
> [2] Scott Fujimoto, et al. Addressing Function Approximation Error in Actor-Critic Methods. ICML, 2018.

---

### Author Response · Authors · 2020-11-20
**Response to all reviewers**

We first summarize our response and the results of additional suggested experiments here. We have responded to the concerns of the reviewers as individual comments below.

All reviewers agree that our idea is interesting and our contributions are valuable to the community. To summarize our contributions: (i) we demonstrate that RERL, a novel framework that combines the strengths of goal-oriented RL and episodic RL, perform well in many complex environments and  (ii) we propose to utilize hindsight and diversity constraints in goal generation, which allows agents to construct and update the regioned memories automatically.

We revised and updated the paper to address the reviewers' comments. In addition,  we conducted more experiments to evaluate the effectiveness of our design within the scope of goal-oriented RL and presented the results in the Experiments section. We want to highlight that, the visualization study on generated goals shows that the exploration is actually promoted by our design. The agent is pulled by its goal and pushed by its anti-goal and goals from other regions, which is in line with our intuitive design motivation.

Here is a list of new experiments:
1. Comparison with [1] (named OPTION in Figure 5(a)).
2. Comparison with [2] (named EMDQN in Figure 5(i)).
3. Ablation Study on goal setting in an environment that is suitable for episodic RL (Atari Game Pong in Figure 5(i)).
4. Visualization study on goal generation (see Figure 6), which is moved from the appendix to the main text.

[1] Jinnai, Yuu, et al. Exploration in reinforcement learning with deep covering options. ICLR, 2019.

[2] Zichuan Lin, et al. Episodic Memory Deep Q-Networks. IJCAI, 2018.

---

### Decision · Program_Chairs · 2021-01-07
**Final Decision**

**Decision:**

Reject

**Comment:**

This paper introduces Regioned Episodic Reinforcement Learning (RERL), which partitions the state space by generating a diverse set of goals and then explores the state space by learning policies that reach those goals. This idea is a combination of episodic memory techniques and “goal-oriented” reinforcement learning.

After the authors’ responses and the discussion phase, all reviewers converged to recommending the rejection of this paper. The main concerns regarding this paper are:

* Presentation. The proposed approach is not that well justified, some of the claims in the paper are quite imprecise, and there’s relevant related work missing.
* Evaluation. The evaluation sometimes feels rushed and is not held in a diverse enough set of tasks, not capturing important properties one would want to capture.

I recommend the authors to pay close attention to presentation, as well as the experiments and analysis in order to make the paper stronger.